# Hippocampal connectivity patterns echo macroscale cortical evolution in the primate brain

Nicole Eichert [1] ✉, Jordan DeKraker[2], Amy F. D. Howard [1], Istvan N. Huszar[1], Silei Zhu[1], Jérôme Sallet[1,3], Karla L. Miller [1], Rogier B. Mars [1,4], Saad Jbabdi[1] & Boris C. Bernhardt [2]

While the hippocampus is key for human cognitive abilities, it is also a phylogenetically old cortex and paradoxically considered evolutionarily preserved. Here, we introduce a comparative framework to quantify preservation and reconfiguration of hippocampal organisation in primate evolution, by analysing the hippocampus as an unfolded cortical surface that is geometrically matched across species. Our findings revealed an overall conservation of hippocampal macro- and micro-structure, which shows anterior-posterior and, perpendicularly, subfield-related organisational axes in both humans and macaques. However, while functional organisation in both species followed an anterior-posterior axis, we observed a marked reconfiguration in the latter across species, which mirrors a rudimentary integration of the default-mode-network in non-human primates. Here we show that microstructurally preserved regions like the hippocampus may still undergo functional reconfiguration in primate evolution, due to their embedding within heteromodal association networks.

The hippocampus is one of the most extensively studied parts of the brain[1]. It is implicated in numerous cognitive and affective processes associated with multiple brain networks, and a model region to examine how neural structure and function covary in space[2–4]. The hippocampus is also markedly affected in multiple common and detrimental indications, including neurodegenerative disorders[5,6], drug-resistant epilepsy[7,8], as well as psychiatric conditions[9,10]. The hippocampal grey matter consists of the archicortex, a phylogenetically old type of cortex, which is considered conserved across mammals[11]. This evolutionary conservation is the basis for translational cross-species frameworks, and we have gained a deep understanding of hippocampal anatomy and function from model species, such as non-human primates[12,13]. Yet, it seems contradictory that the hippocampus supports many functions sometimes considered unique to humans, such

as autobiographical memory[14], future thinking[15], and self-perception[16]. This apparent paradox can be resolved by two potential explanations: Firstly, it is possible that species differences in primate hippocampal structure have been overlooked as evolutionary diversification of hippocampal anatomy has rarely been studied (but see[17]). Therefore, we need quantitative frameworks that go beyond measuring regional brain volumes to compare species. Or, secondly, the integration of the hippocampus within the rest of the brain has undergone fundamental reconfiguration since the last common ancestor between humans and monkeys. Species-specific specialisations in subcortical structures such as the striatum[18,19] or the amygdala[20] support the second hypothesis.

The functional embedding of the multiple subdivisions of the hippocampus with the rest of the brain is diverse. On the one hand, the hippocampus is directly connected to limbic and paralimbic

[1]Wellcome Centre for Integrative Neuroimaging, Centre for Functional MRI of the Brain (FMRIB), Nuffield Department of Clinical Neurosciences, John Radcliffe Hospital, University of Oxford, Oxford, UK. [2]Department of Neurology and Neurosurgery, Montreal Neurological Institute and Hospital, McGill University, Montreal, Canada. [3]INSERM U1208 Stem Cell and Brain Research Institute, Univ Lyon, Bron, France. [4]Donders Institute for Brain, Cognition and Behaviour, Radboud University Nijmegen, Nijmegen, The Netherlands. ✉e-mail: nicole.eichert@ndcn.ox.ac.uk

structures, such as the amygdala and cingulate cortex, which host some of the most preserved circuits of the brain[21]. On the other hand, it is closely linked to heteromodal regions in the neocortex, including the dorsal-lateral prefrontal and parietal cortex, and it is considered part of the classical default-mode-network (DMN)[22–24]. The definition of the DMN across the primate lineage, however, is a matter of debate and it has been suggested that it forms two sub-networks in non-human primates[25]. Any evolutionary reconfiguration of the hippocampus is likely related to its functional embedding with the DMN[25,26], but quantitative evidence for this theory is sparse[27]. Mapping cross-species differences in functional anatomy of the hippocampus is, therefore, critical both for understanding the origins of human cognition, and also to identify the limitations of model species for studying human disorders such as schizophrenia[28].

Here, we set out to test these hypotheses in a study that compares macaques and humans. We devised a computational approach to interrogate cross-species differences in hippocampal anatomy, microstructure, and functional organisation in a common reference frame[29]. We capitalise on recent computational approaches to analytically unfold the hippocampal formation and to derive a surface-based coordinate system[30]. This topological framework maps the hippocampus intrinsic long (anterior–posterior) and short (distal–proximal) axes, thus respecting the sheet-like anatomy of the hippocampus. Representing the cortex in a surface-based coordinate system has previously proven to advance efforts in brain mapping[31]. Specifically for the hippocampus, such a data-driven estimation of hippocampal coordinates allows us to establish subregional correspondence in a cross-species setting, independent of a specific definition of hippocampal regions or subfields. It nevertheless allows for the integration of multi-modal data ranging from high-resolution histological data to in-vivo functional MRI in a shared framework.

Leveraging this common space, we aimed to characterise the subregional microstructural organisation of the hippocampus. This work represents the first integration of microstructural features derived from a recently developed multimodal and multiscale macaque atlas, including whole-brain *post-mortem* histological information[32] with microstructural features of the human hippocampus[33]. Then, based on the detailed analytical unfolding of hippocampal anatomy and microstructure, we characterised the spatial axes of hippocampal function and its embedding within macroscale functional systems. This then allowed us to test whether spatial axes of hippocampal function underwent a diversification in humans relative to macaques and whether this diversification coincided with the reconfiguration of cortex-wide functional systems.

We introduce a comparative space that represents the hippocampus as an unfolded surface, allowing for meaningful comparison across species. Notably, multimodal data (microscopy, structural and functional MRI) can be integrated within the same space, enabling simultaneous study of geometry, microstructure, and function. We use this framework to study hippocampal differences between humans and macaques. We hypothesise that these species have similar hippocampal microstructure and that cross-species differences relate to hippocampal connectivity with higher-order functional networks. Our methodology lays the groundwork for a multi-modal comparative approach, where similar methods can be used to integrate information across modalities and translate information across species, to elucidate the evolutionary trajectories and underlying mechanisms shaping human cognitive abilities.

## Results

### The hippocampal microstructure is conserved

We adapted a recently developed method for analytically unfolding the hippocampus (hippunfold[30]), to the macaque brain, as demonstrated on a template MRI scan based on 10 ex vivo macaques[34] (Fig. 1A) for comparison to the human. The 3D surface reconstruction revealed the characteristic seahorse shape of the hippocampus (Fig. 1B). The unfolded flatmap space is defined based on intrinsic coordinates of the hippocampus, ranging from posterior to anterior (from tail to body and head) and from distal to proximal. Proximal in this context refers to the structures closer to the dentate gyrus, whilst distal refers to those regions closer to the subiculum[35]. Note this DG-centric terminology differs from the terminology used in previous related hippunfold-publications, which used terms relative to the neocortex. The flatmap covers all subfield-related parts of the hippocampus in full, and the resulting surface expansion map is shown in Supplemental Information, Fig. S1A. The tip of the hippocampal tail is represented at the uppermost edge of the flatmap. In 3D volumetric space, however, the tail and head region curl toward the body of the hippocampus as shown in Supplemental Information, Fig. S1B. All hippocampal flatmaps in the remainder of the paper are shown according to this orientation. An animated visualisation of the macaque hippocampal subfields is provided in Supplementary Video 1, where we show a 3D rendering alongside the 2D flatmap and the histology slices, with the virtual cutting plane moving from posterior to anterior. The hippocampal surface space reconstructed with hippunfold matched geometrically equivalent points across macaque and human. This is demonstrated in Fig. 1C, which displays geometric indices of the macaque hippocampus next to those of the human.

To investigate the hippocampal microstructure in the macaque, we reconstructed the hippocampal surface in a single macaque scan from the BigMac dataset, an open resource combining multi-contrast and ultra-high-resolution microscopy and MRI in a single macaque brain[36] (Fig. 1D). We mapped histological measures (Cresyl violet stain for Nissl bodies and Gallyas Silver stain for myelin), manual labels of hippocampal subfields and three MRI metrics (fractional anisotropy [FA], mean diffusivity [MD], multi-gradient-echo intensity [MGE]) to the hippocampal surface. The microstructural mappings demonstrated that hippocampal microstructure varies primarily along the distal-proximal axis and the characteristic spiral configuration of subfields was represented as a sequence of vertical subfields in the flatmap (Fig. 1E). These variations follow common patterns, for example, the subicular complex had higher intensity in the Gallyas stain and lower MGE intensities compared to CA1, reflecting the higher amount of axons relative to cell bodies, in line with previous reports in the human[30,37]. The unfolding algorithm introduced some mild distortions evident by a twist in the most anterior third of the subfield map (Fig. 1E). This effect was independently observed in both species. We compared the macaque hippocampal map to that from the human BigBrain, and the overall pattern was highly similar (similarity metric of 0.95 and 0.93 for left and right hippocampus, across all subfields, Supplemental Information, Fig. S1F). Subtle differences in the relative extent of hippocampal subfields, however, as quantified using pairwise comparisons were also observed (Supplemental Information, Fig. S1G). For example, CA2 and CA3/4 are relatively expanded in humans. For validation, we repeated hippocampal mapping of the MGE contrast in two additional ex vivo scanned macaque brains and observed consistent results (Supplemental Information, Fig. S1E).

Taken together, our hippocampal unfolding and microstructural mapping demonstrated that anatomy and microstructure are overall preserved in both humans and non-human primates, despite subtle changes in global shape and subfield proportions.

### The functional embedding of the hippocampus is diverse

We continued to study the functional anatomy of the hippocampus using resting-state functional MRI (rs-fMRI) data from 10 adult individuals in both species. Rs-fMRI has been widely used to investigate the intrinsic functional brain organisation in both species[38], and the comparability of functional networks between awake and lightly anaesthetised states has been firmly established[39–41]. Image pre-processing and analysis were equivalent in both species and the macaque fMRI

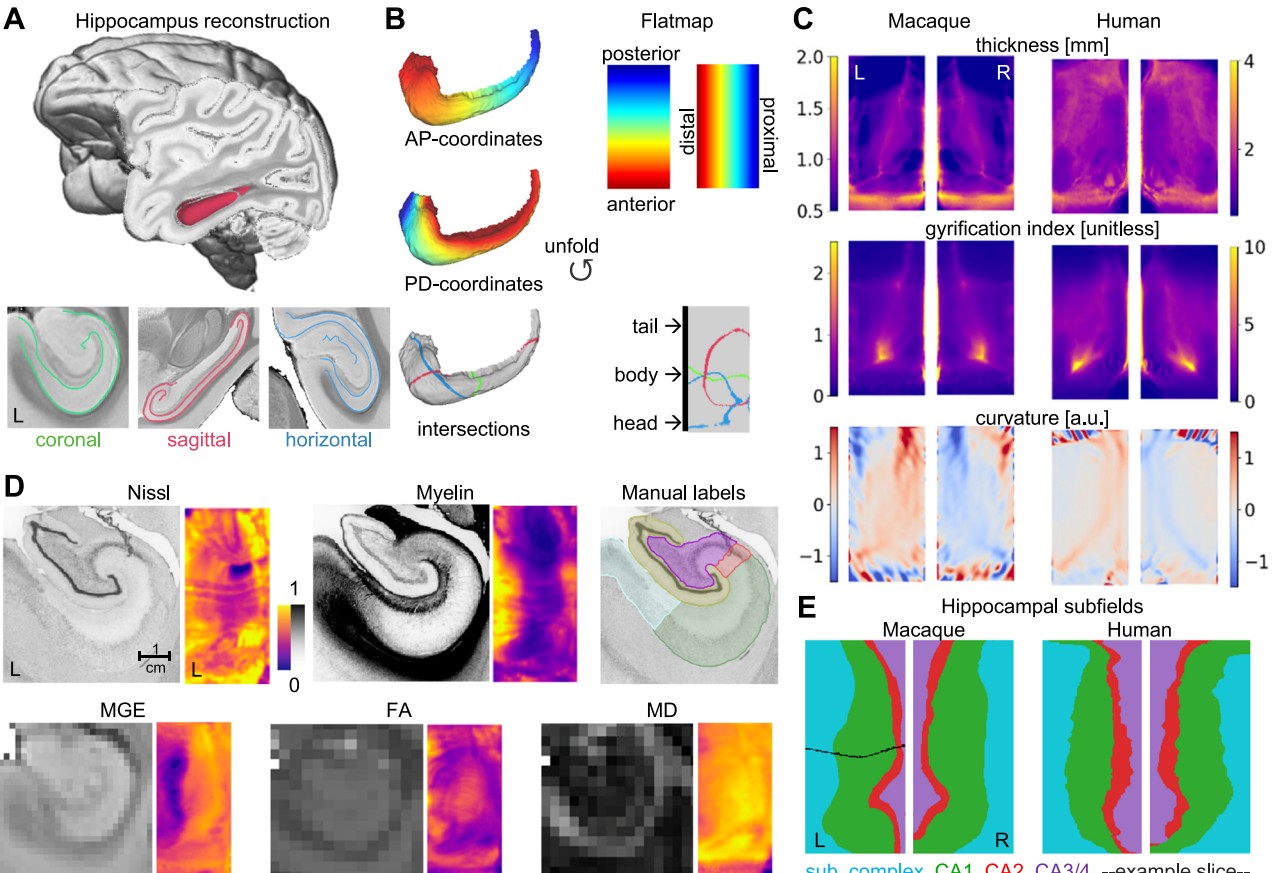

**Fig. 1 | The hippocampal microstructure is conserved. A** The hippocampal surface was reconstructed in a template MRI scan. **B** Left−Hippocampal surface as 3D reconstruction and as 2D flatmap. Right−Shown are the intrinsic hippocampal coordinates at the top and the intersections with the planes in (**A**) at the bottom. **C** Geometric indices in hippocampal flatmap space for macaque and human. **D** Histological and MRI metrics from BigMac. All histology slices and MRI scans were acquired in the same individual macaque brain. For each modality we show an example coronal section and the whole hippocampus mapped to the flatmap. **E** Macaque (BigMac) hippocampal subfields and human (BigBrain) subfields. Black line: Intersection with the example histology slice shown in (**D**). AP anterior−posterior, PD proximal−distal, MGE multi-gradient-echo, FA fractional anisotropy, MD mean diffusivity, sub. complex subicular complex, a.u. arbitrary unit.

data are of higher tSNR than most previously used datasets[42], altogether suggesting that the data in both species allowed for a quantitative comparison.

Application of non-linear dimensionality reduction techniques[43] to hippocampus-to-cortex connectivity matrices provided spatial maps of connectivity variations, also referred to as Connectivity Gradients[22,44,45], of the hippocampus (Fig. 2A). The first component revealed a pronounced anterior-posterior axis in both species (Fig. 2B). Later components (2nd in the human and 6th in the macaque) showed further differentiation along the distal−proximal axis (Supplemental Information, Fig. S2A). These distal−proximal gradients showed variations that appeared to differ across hippocampal subfields. This finding was consistent with recent reports on functional connectivity differences along the anterior−posterior axis of hippocampal subfields obtained using high-resolution precision scanning protocols[46]. However, due to limits in spatial resolution of the fMRI data available for our study, these variations were not further investigated.

Next, we characterised the functional connectivity of the hippocampus with the rest of the cortex (Fig. 2C). To this end, we performed simultaneous decomposition of cortex-to-hippocampus connectivity data in both species. This joint connectivity gradient mapping approach provided us with homologous spatial maps of connectivity variation across the cortex. One apex of the 1st component (warm colours in Fig. 2C) recovered a network of homologous areas in both species, including posterior cingulate/precuneus, lateral temporal

lobe, supramarginal gyrus, ventro-medial and dorso-lateral prefrontal cortex. The regions within this network were all highly connected to the hippocampus (Supplemental Information, Fig. S3A). In both species, hippocampal connectivity was also observed in the inferior temporal gyrus and occipital lobe, but to a lower extent. Some species differences in this network, however, were found as well. For example, in macaques, the supramarginal gyrus and somatosensory cortex were less differentiated, compared to the human. Overall, however, our analyses uncovered a homologous brain network in both species, defined by similar connectivity profiles to the hippocampus.

In humans, the resulting network from our hippocampal analysis was reminiscent of a default mode network (DMN). This finding is in line with our expectation, as the hippocampus is a well-established node of the human DMN[47]. The macaque network that emerged from the same analysis can be thought to represent the macaque homologue of the DMN. However, previous definitions of the macaque DMN using, for example, independent component analysis decomposition techniques suggested that the macaque DMN consists of two sub-networks[25] or only a partial network[26,48]. To demonstrate that the homologous macaque DMN, as we defined it using hippocampal connectivity, in fact, comprised two distinct cortical networks, we conducted additional analyses.

First, we performed dimensionality reduction of cortico-cortical connectivity matrices in both species and compared these to the hippocampal network from the joint connectivity gradient analysis

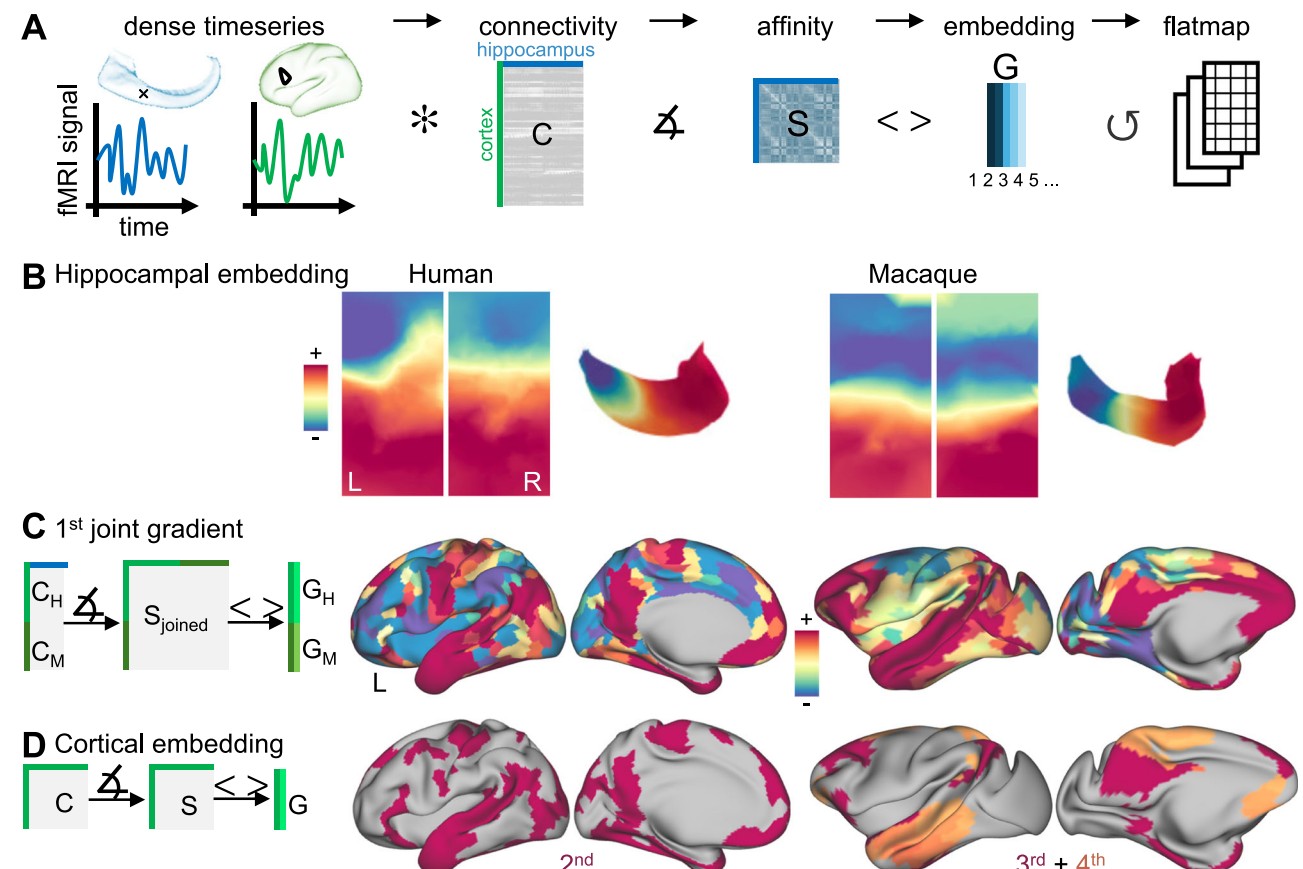

**Fig. 2 | The functional embedding of the hippocampus is diverse. A** Workflow to perform hippocampal gradient embedding. A dense connectivity matrix (C) is constructed from vertex-wise hippocampal and parcellated cortical resting-state data. After applying a similarity operation, the square matrix (S) is decomposed into gradient maps (G). **B** First hippocampal gradient in humans and macaques shown as flatmap and on the left hippocampal surface reconstruction. **C** Left—A joint cross-species cortico-hippocampal gradient (G) was obtained by concatenating the human cortico-hippocampal connectivity matrix ($C_H$) with that of the macaque ($C_M$) followed by an affinity operation (S) and gradient embedding (G). Right—1st joint cross-species gradient in human ($G_H$) and macaque ($G_M$). **D** Left—In each species separately, cortico-cortical gradients were computed. Right—Thresholded maps of cortical gradients that match the joint cross-species gradient. Symbols: ✳ = cross-correlation of timeseries, ⊿ = affinity operation, < > = non-linear gradient embedding, ↻ = hippocampal unfolding, + (−) = positive (negative) apex of embedding dimension.

above. All cortical networks from both species are shown in the Supplemental Information, Fig. S2B. As expected in humans, one cortical network map matched the joint gradient best (regularised regression analysis, coefficient = 0.76, $r^2$ = 0.55, $p_{corrected}$ < 0.001). The best fit for the joint gradient in the macaque was found with the 4th cortical embedding map (coefficient = 0.46, $r^2$ = 0.16, $p_{corrected}$ < 0.001), closely followed by the 3rd (coefficient = 0.21, $r^2$ = 0.06, $p_{corrected}$ < 0.05). Successive Dice overlap analysis confirmed that a combination of map 3 and 4 in the macaque matches the joint gradient best (Dice = 0.54, at threshold 75%). This analysis confirmed that the macaque homologue of the DMN as defined by hippocampal embedding was best explained by a combination of two distinct cortical networks: The 4th cortical embedding map recovered the lateral temporal and medial frontal DMN nodes, while the 3rd gradient recovered the inferior parietal, the dorso-lateral prefrontal and the precuneus DMN nodes (Fig. 2D). Precisely these two subnetworks have previously been suggested to form the DMN in non-human primates[25].

To extend the cross-species comparison to a whole-brain level, we finally computed a vertex-wise homology index based on the connectivity profiles with the hippocampus in both species[49,50]. This analysis confirmed that somatomotor and limbic networks were most conserved across species, whilst higher-order networks such as the DMN and multiple-demand network, as defined by the discrete Yeo human network parcellation[51], were more strongly reconfigured (Supplemental Information, Fig. S3C).

Our functional analysis showed that the hippocampus is embedded with homologous large-scale functional networks in both species with strongest involvement of the DMN. Whilst the DMN-homologue in the macaque formed two distinct sub-networks on the cortical level, the full macaque DMN functionally interacts with the hippocampus. Taken together, we showed that the short hippocampal axis primarily captures variations in microstructure, whilst the long axis primarily characterises variations in functional organisation.

### Cortical embedding of the hippocampus reflects evolutionary reorganisation

Finally, we studied how the functional topography of the hippocampus is reflected in macroscale cortical networks and explicitly investigated how the two established hippocampal axes map onto the cortex. We used dual-regression[52,53] to determine the individual contribution of the two orthogonal hippocampal gradients to cortical connectivity (Fig. 3A).

The cortical reflection of the long-axis gradient showed that the anterior part of the hippocampus notably mediates connectivity with the DMN nodes in both species (warm colours in top Fig. 3A: angular gyrus, middle temporal lobe, posterior cingulate/precuneus, dorsolateral frontal and anterior medial as well as orbital frontal cortex). Preferential connectivity with the posterior hippocampus, however, displayed a pattern previously described as multiple-demands-

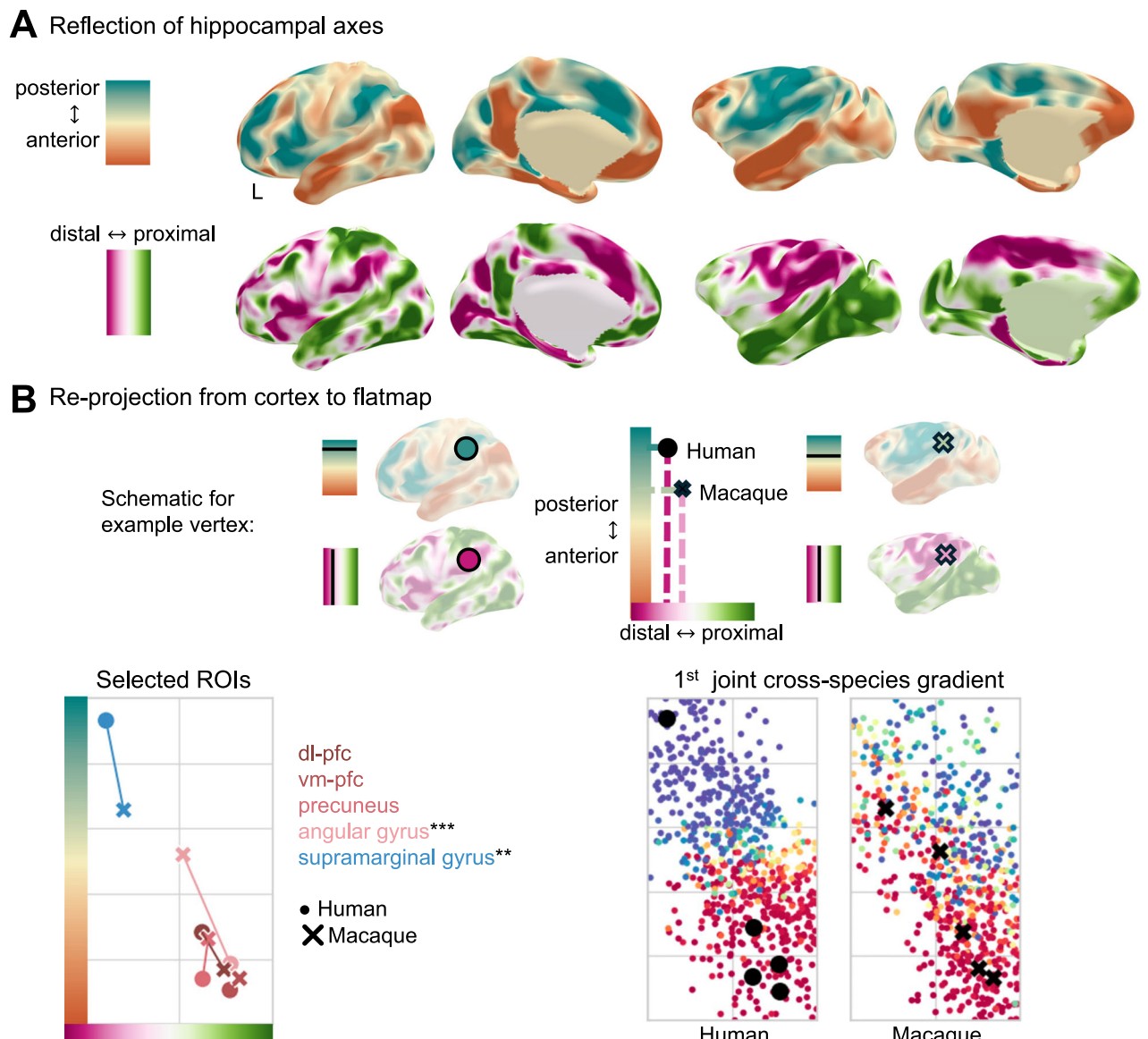

**Fig. 3 | Cortical embedding of the hippocampus reflects evolutionary reorganisation. A** Cortical reflection of the hippocampal axes derived using dual regression. The colours in the cortical maps (right) represent differential connectivity to different parts of the hippocampus (left). Colormaps are matched for both species (10–90%). **B** Top—Schematic diagram to demonstrate how an example cortical vertex is remapped to the 2D flatmap space based on its connectivity in (**A**). The aspect ratio of the coordinate system reflects that of the hippocampal space. Bottom left—Five brain areas in both species mapped onto the 2D space. The standard deviations are shown in Fig. S3E. We tested the difference between human and macaque using two-sample two-sided Kolmogorov–Smirnov test (***$p = 5.57 \times 10^5$, **$p = 0.0011$, $n = 10$ individual subjects per species, corrected for the number of comparisons performed). Bottom right—1st joint cross-species gradient (Fig. 2C) represented in a 2D space. Black data points are the selected brain areas shown in the right panel. dl-pfc dorso-lateral prefrontal cortex, vm-pfc ventromedial prefrontal cortex.

network[54]. A notable cross-species difference, however, was observed in the inferior parietal lobe, which showed a clear dissociation along the anterior-posterior axis in the human, but little differentiation in the macaque. Connectivity maps of spatially distinct hippocampal segments confirmed these patterns (Supplemental Information, Fig. S3B).

The cortical reflection of the distal-proximal axis revealed a more distributed pattern. Connectivity to DMN nodes, but also to higher-order visual areas and ventral premotor cortex were mediated by more distal, i.e., subicular, parts of the hippocampus. Again, the inferior parietal lobe showed a clear distinction within this axis in the human, but not in the macaque.

The differential connectivity along the two hippocampal axes allowed us to map cortical brain areas back into a two-dimensional coordinate system, which in turn represented the intrinsic hippocampal space (Fig. 3B). We leveraged this visualisation to map selected brain areas: Four nodes of the DMN (ROIs with warm colours) and the supramarginal gyrus (ROIs with cold colours). This visualisation highlighted that the DMN nodes from both species map onto similar locations in the shared space. The two nodes of the inferior parietal lobe (angular and supramarginal gyrus) exhibited a strong dissociation in humans falling onto opposite corners of this space. In the macaque, however, both inferior parietal nodes showed much reduced differentiation along both axes. We further reprojected the joint cross-species gradient from above (Fig. 2C) to this space, demonstrating that connectivity with this hippocampal network is mediated by anterior and distal parts of the hippocampus in both species.

Taken together, these results demonstrated that the DMN nodes exhibit differentiated connectivity with the hippocampus reflecting the intrinsic topography. The macaque inferior parietal lobe reflects incomplete integration of the DMN in the non-human primate.

## Discussion

Studying the hippocampus, a key interface of paralimbic and hetero-modal association systems, provides important insights into primate evolution and conservation[11]. This paper devised a computational comparative framework for studying the evolutionary reconfiguration of hippocampal microstructure, anatomy, and integration into large-scale systems. To this end, we successfully unfolded the non-human primate hippocampus using a recently developed analytical tool[30]. The hippocampal 2D surface is geometrically matched across species and, thereby, represents a common space for comparative analyses[29]. First of all, we found that the microstructural blueprint of the hippocampus, its principal functional axes in anterior–posterior direction, and cortical network embeddings are overall conserved across the two species. However, hippocampal embeddings within macroscale functional networks also reflected cross-species differences, particularly with respect to its functional connectivity within the default-mode network (DMN). Specifically, the inferior parietal lobe in the macaque mirrors an incomplete integration of the DMN compared to the human. Our findings, thus, suggest that the human DMN has expanded and further integrated in the human lineage, harnessing the hippocampal micro-circuit in a potentially uniquely human way. Altogether, these adaptations may form a basis for specialised human brain function spanning a wide range of cognitive processes. Expanding our framework to include a larger set of primate species will allow us to determine if these effects are truly unique to humans, or gradually evolved across primates. While the hippocampus' structural organisation is largely conserved, its functional connectivity has evolved. Our findings demonstrate that even structurally preserved regions like the hippocampus can undergo functional adaptations due to their connections with higher-order networks.

We leveraged two unique ultra-high-resolution multiscale data-bases from the macaque and human for a cross-scale and cross-species comparison. This combination of resources allowed us to show that hippocampal macro- and microstructure are overall conserved across species, a long-standing evolutionary hypothesis that was lacking spatially resolved quantification to date. The histological mapping further served as a microstructural validation of our hippocampal surface and flatmap, confirming that it represents a comparative space across primates. Our findings, overall, showed that both species have a comparable long axis, comprising head, body and tail regions that are characterised by a canonical sequence of subfields. This demonstrated that microstructural patterns and gradients in the hippocampus are overall conserved across species, suggesting that the basic micro-circuit and hippocampal computation remained largely unchanged over evolutionary time. Despite the pronounced similarities in subfield mapping across species, however, we also showed that our framework can detect nuanced differences. For example, our findings suggested a relative expansion of the CA2 subfield in the human, an effect that may be compatible with the hypothesised functional specialisation of CA2 for social memory[55] and territorial behaviour[56], underpinned by the subfields distinct cytoarchitecture[54] and genetic profile[57].

Although findings on potential cross-species microstructural differences require further validation in a larger histological sample to discern inter-species from inter-individual differences, the metho-dology we introduced here offers a scalable framework to allow for microstructural and subregional comparisons between humans and non-human primates. Beyond the value for fundamental neuroscientific enquiry, as carried out in the current study, this approach may also be beneficial when translating findings from animal models to patient groups in a preclinical/clinical context. Methods to automatise hippocampal subfield segmentation[30,58,59], and to enhance cross-modal and inter-individual registrations[30] as well as multi-scale contextualisation[60], are already well established in humans, and cur-rent efforts to adapt them to non-human primate brains will facilitate that process[60].

Building upon the microstructural analysis, our resting-state fMRI results demonstrated that the functional differentiation along the hippocampal long axis is also largely preserved across the two species. The cortical embedding of the hippocampus recovered a homologous DMN in both species comprising dorso-lateral frontal, inferior-parietal, anterior-temporal, as well as fronto- and posterior-medial regions. Importantly, this homologous DMN was defined based on the joint embedding of hippocampal connectivity and did not require the specification of homologous cortical regions-of-interest. Because of the data-driven nature, anchored only by the hippocampus itself, our approach is readily applicable to all mammalian species and represents an important further development to previous remapping approaches[49,50]. General limitations of cross-species comparisons using resting-state fMRI were mitigated by using a macaque dataset with high temporal signal-to-noise ratio, long scanning time, light anaesthesia and by using an equivalent HCP preprocessing pipeline in both species. Reproducing our human findings with different anaes-thesia protocols would be a worthwhile replication.

Despite the overall similarity of functional networks, we also observed notable species differences in functional connectivity. In particular, we showed that the DMN, known to act as a major inte-grated network in humans[24,47,61], constitutes two subcomponents in the macaque when studied on the cortical level only. Various previous definitions of the non-human primate DMN homologue referred only to one of the two subnetworks[48] or described them as two distinct subnetworks[25]. Our analysis suggests, however, that all conventional DMN nodes are present as precursors in the macaque. The critical species difference we propose is that macaques exhibit an incomplete integration of the DMN, whilst humans have a fully integrated network. The increase in network integration and connectivity in the human lineage mirrors the increase in DMN integration during typical human brain development, seen both functionally[62,63] and structurally[64–66]. Our findings thus provide new evidence to support the hypothesis that the computational landscape of the mature adult brain is fundamen-tally linked to the development of long-range connections and the development of the DMN. The cortical embedding of the two hippocampal axes revealed that the inferior parietal lobe in the macaque specifically reflects this incomplete network integration. Our results suggest that parietal connectivity to the temporal lobe and posterior medial cortex are reduced in macaques, which is in line with previous comparative literature on structural[67,68] and functional connectivity[38,49]. In addition to changes to connectivity, the parietal lobe is one of the hotspots of cortical expansion, which is supported by uniquely human genetics[69]. These adaptations underpin the specia-lised role of the parietal lobe in social cognition[70] and theory of mind[71] in humans. Our findings, therefore, reconcile and extend previous controversies about the evolution of the DMN[26] and the functional significance of DMN integration for mature human brain function.

A wealth of data from multiple species and modalities suggests that hippocampal organisation can be conceptualised along two quasi-orthogonal axes[22,54,60]. The spatial gradients we proposed in this paper, therefore, serve as a meaningful coordinate system to study pre-servation and innovation in brain evolution. The short distal-proximal axis captures microstructural variations traditionally described as discrete subfields[54,72,73]. These structural variations have been well established in a range of mammalian species, and they are the neural substrate for a specialised circuitry that forms the basis of hippo-campal function[74]. Indeed, our histological analysis demonstrated that the subregional organisation and, therefore, the canonical hippo-campal microcircuit along the distal–proximal axis is largely conserved

across the primate lineage. The spatial topology of the hippocampus, however, is also characterised by variations along the long anterior-posterior axis[4], which recapitulates the segmental head-body-tail anatomical arrangement. Notably, spatial patterns in gene expression[75], receptorarchitecture[76] and hippocampal function[77] have been shown to closely follow this long axis in the human. Our results provide an empirical demonstration of a long-axis functional differentiation in non-human primates (see also[78]) and demonstrate direct correspondence to the human. The diverse brain-wide connectivity of the hippocampus is thought to mirror the functional substrates of hippocampal involvement in various cognitive and behavioural domains[79]. In line with this theory, our analysis demonstrated that the functional connectivity of the hippocampus is diverse and varies along both intrinsic axes. Overlapping and covarying axes of microstructure, connectivity, and molecular profiles together can possibly explain how the hippocampus can be engaged in many different brain functions[3] ranging from relational memory across spatial, episodic and semantic contexts[80–83] to other domains including emotional reactivity[22,84] and stress[85]. While we found that the fundamental microstructure of the hippocampus is overall preserved, cross-species differences in the functional embedding of the hippocampus are possibly mirrored in nuanced species differences on the level of gene expression or receptorarchitecture[86].

Interpreting our findings in the wider context of evolutionary adaptations ultimately requires an expansion of our framework towards more primate species and other mammals. Thanks to advances in the field of MRI, such a phylogenetic approach is becoming increasingly possible for whole-brain characteristics[87], for example, structural connectivity[88,89], cortical folding[90], or brain function[91]. While the current study relied on a manual and, therefore, imperfect macaque MRI tissue segmentation, ongoing developments of the software will further refine the level of detail and range of input modalities and species towards the observer-independent and automatic unfolding of the primate hippocampus. In this context, open data sharing projects such as the BigMac dataset[32] used in this study, but also initiatives such as PRIME-DRE[92] may make an invaluable contribution, as they will allow for the aggregation of a diverse set of data in non-human primates, and their dissemination to a wide range of researchers. They also pave the way for functional connectivity MRI studies[25] to overcome the current limitation of small numbers of species.

Taken together, our work provides support for long-standing theories of brain evolution. The multisynaptic pathway of the hippocampus likely emerged in early mammals to promote survival in complex environments via spatial navigation and pattern separation[80,93]. According to existing theories, rodents leverage this basic computation for integration of proximal and spatial cues[94], whilst primates repurpose the circuit to integrate visual and abstract cues[81] or values[95]. This integration relies on extended connectivity with other brain regions and eventually gives rise to higher-order cognitive abilities, such as episodic memory[96] and social cognition[97]. The hippocampal function, therefore, essentially co-evolved with heteromodal systems such as the DMN and, at the same time, maintained its capacity to integrate sensory processing streams[98,99]. The preservation of a successful microcircuit and simultaneous gain of function by virtue of specialised cortical connectivity makes the hippocampus a prototypical region to understand human brain evolution. This interpretation of our findings fits in well with the growing body of literature on evolution, demonstrating that human brain evolution is a nuanced process that goes well beyond brain expansion. On the level of cortical brain areas, cross-species work across the mammalian family has demonstrated that multiple forms of adaptations can overlap and interact, such as the relative size of cortical fields and local changes to connectivity[29,91]. This concept extends to the brain network level as previously demonstrated for the language system in the brain[100]. Regional expansions of the cortex, for example, cannot explain the changes to long-range white

matter tracts across primates[101]. These regional expansions are most pronounced in association cortex and mirror patterns of brain expansion during human development[102]. It has been suggested that these rapid expansions free up portions of the cortex that become untethered or from early molecular constraints imposed by conserved brain anchor regions. The untethering of these regions allows for the development of more dense connections within and between brain networks[103]. Our results align well with the untethering hypothesis, where the hippocampus forms a conserved anchor for the primate DMN, which increasingly expanded and integrated in humans thus endowing the hippocampus with increased functionality.

In conclusion, we developed a comparative framework to study the hippocampus across species. Our in-depth study of the primate brain integrated ultra-high-resolution assessments of hippocampal microstructure with advanced decompositions of its functional network embedding and demonstrated how conserved brain regions can functionally adapt through interactions with advanced networks. We anticipate this paper to be the starting point for the next generation of comparative studies, to unlock a deeper understanding of the evolution of our own cognitive abilities.

## Methods
### Human−MRI data
Human MRI data were provided by the Human Connectome Project, WU-Minn Consortium (Principal Investigators: David Van Essen and Kamil Ugurbil; 1U54MH091657) funded by the 16 NIH Institutes and Centers that support the NIH Blueprint for Neuroscience Research; and by the McDonnell Center for Systems Neuroscience at Washington University. We accessed minimally pre-processed structural and rs-fMRI data in fsLR_32k space of 10 subjects from the 3 T 1200 Subjects Data Release[104] (4 females, mean age 30.4 ± 3.2 years) to match the sample size in the macaque. The subjects were randomly selected. Data from one rs-fMRI run were accessed for each subject, acquired with the following parameters: 2 mm³ isotropic resolution, TR = 0.72 s, 1200 volumes, ~14 min.

### Human−histological data
Previously established human histological subfield mapping was based on a single post-mortem sample from a 65-year-old male donor, BigBrain[33]. This human hippocampal subfield map was manually defined based on histology data mapped to 3D. We accessed only the surface map of discrete hippocampal subfields, which are provided via open-access by a previous study[105].

### Macaque−post-mortem MRI data
A high-resolution (0.15 mm³) rhesus macaque (*Macaca mulatta*) structural gradient-echo macaque template based on 10 individuals was accessed from the CIVM Macaque Brain Atlas[34]. Furthermore, we accessed 7 T post-mortem whole brain data from a male adult rhesus macaque, the BigMac Dataset[36], provided openly accessible by the Oxford Digital Brain Bank (open.win.ox.ac.uk/DigitalBrainBank)[106]. Full details of the BigMac data acquisition and preprocessing are provided in the original publication[36]. Specifically, we accessed the multi-gradient-echo (MGE) high-resolution structural scan (0.3 mm³) as well as pre-processed fractional anisotropy and mean diffusivity data from the *b* = 4k diffusion acquisition (0.6 mm³). Histological data (see below) from the same individual was also accessed. Post-mortem MGE scans were acquired using the same sequence in two additional macaque brains.

### Macaque−in vivo MRI data
All macaque in vivo data were acquired for previous studies[38,40] and reanalysed for the purpose of the present paper. Resting-state functional and in vivo structural MRI data were obtained from 10 rhesus macaques (*Macaca mulatta*, 1 female, mean age at scan 7.2 ± 2.5 years).

Details of the scanning protocol and physiological monitoring are described in a previous publication[38]. In short, the animals were scanned in a 3 T scanner under light isoflurane anaesthesia whilst placed in an MRI-compatible stereotactic frame or resting on a custom-made mouth mould. In short, BOLD fMRI was acquired for 1600 volumes (-1 h) with the following parameters: 1.5 mm³ spatial resolution, TR = 2280 ms, TE = 30 ms. Structural scans using a T1-weighted MPRAGE sequence were acquired at 0.5 mm³ during the same scanning session.

## Macaque−histological data

We accessed microscopy data from the BigMac dataset, specifically the Cresyl Violet and the Gallyas Silver stain sections, which had been digitally downsampled to a resolution of 40 μm. We also accessed previously generated registrations for each histology slice to the MGE volume. These registrations were derived using FSL's TIRL v.3.1[107,108] using a sequence of linear and non-linear transforms and a modality-independent cost function.

## Ethics statement

All HCP scanning protocols were approved by the local Institutional Review Board at Washington University in St. Louis. All subjects provided informed consent prior to participating in the study. The donor for the post-mortem BigBrain sample is not personally identifiable and gave written informed consent for the general use of post-mortem tissue used in this study for aims of research and education. The usage of the post-mortem material is covered by a vote of the ethics committee of the medical faculty of the Heinrich Heine University Düsseldorf (#4863). All experimental procedures in macaques were performed in compliance with the United Kingdom Animals (Scientific Procedures) Act of 1986. A Home Office (UK) Project License, obtained after review by the University of Oxford Animal Care and Ethical Review Committee, licensed all procedures. The housing and husbandry followed the guidelines of the European Directive (2010/63/EU) for the care and use of laboratory animals. The 3Rs principles of compliance and assessment were conducted by the UK National Centre for 3Rs (NC3Rs).

## Overview of hippocampal unfolding approach

First, we conducted an MRI tissue segmentation to identify robust landmarks of the hippocampus and surroundings. In the macaque, we defined the segmentation manually, whilst in the human, the corresponding segmentation was derived automatically. Next, the software tool hippunfold[30] was used to estimate hippocampal coordinates along the short and long hippocampal axis based on this segmentation. The tool further reconstructs the hippocampal surface and computes a coordinate transformation to achieve analytical unfolding or flattening of the hippocampus. This ultimately results in a surface-based coordinate system that is matched across the two species. Hippocampal subfield labels were manually defined in the macaque histology data, then translated to MRI space and finally sampled along the hippocampal surface, resulting in a 2D map. In humans, the corresponding map of subfields was accessed from a previous study[105].

## Hippunfold

To reconstruct the hippocampal surface, we used a recently developed computational tool, hippunfold v.0.3. Hippunfold requires tissue segmentation to define unfolded coordinate boundaries of the hippocampal surface. Note, that this tissue segmentation should not be confused with the definition of hippocampal subfield labels, which is based on histology, as described below. In humans, the segmentation can be derived automatically by hippunfold using a convolutional neural network, but an adaptation for the macaque brain required us to manually segment a set of hippocampus and surrounding structures. We developed the macaque tissue segmentation in a high-

resolution (0.15 mm³) gradient-echo MRI scan from the CIVM database. The following labels were manually segmented in ITK-SNAP v.3.8.0[109]: The subfield-related regions of the hippocampus (Cornu Ammonis, CA), the dentate gyrus (DG), the hippocampal dark bank, the grey matter of the temporal lobe directly adjacent to the hippocampus, the uncus, the hippocampal-amygdalar transition area, and indusium griseum. The dark band label covers a heterogenous set of structures, including parts of the archicortical strata radiatum, lacunosum and moleculare, but also other axons and incidental structures, such as residual cerebrospinal fluid and blood vessels in the hippocampal sulcus. We followed the tissue segmentation protocol developed for the human[110], with minor notable differences: (i) the macaque hippocampus was less gyrified and so the volumetric segmentation was overall simpler, (ii) the uncus of the hippocampus was smaller in the macaque, but still showed the same critical termination on the amygdala that allows for unfolding, and (iii) the boundary between subiculum and medial temporal lobe neocortex was shifted laterally in the macaque compared to the human, making the macaque hippocampus smaller, to accommodate the darker intensity of the parasubiculum which was visibly shifted laterally in macaques compared to humans. For further explanations for each MRI label, its relations to histology and the criteria used for manual definition, the Reader is referred to the original human protocol[110].

The tissue segmentation was developed in the left hemisphere, then initialised in the right hemisphere by non-linear registration using ANTs v.2.2.0 QuickSyN tool[111] and manually corrected in the right hemisphere. Using the segmentations as input, hippunfold was run using default settings, and two surface meshes at a resolution of 419 vertices per hemisphere (low-resolution mesh, Fig. S1A) and 7262 vertices (high-resolution mesh) were obtained. In addition to the outer, inner, and mid-thickness surface, we obtained in total 6 equi-volumetric surfaces across the hippocampal depth. The hippocampal flatmap space is derived by hippunfold by estimating the 2D Laplacian coordinates of the archicortical sheet. An expansion showing the distortions of the surface is provided in Fig. S1A. Hippunfold automatically computes vertex-wise measures of hippocampal thickness, gyrification and curvature. Furthermore, the previously labelled human hippocampal subfield labels from BigBrain are provided as a categorical surface map.

For resting-state analysis (*see below*) the hippocampal surfaces from the CIVM template were non-linearly transferred to the Yerkes-19 macaque template space[112,113]. In the MNI template brain, hippunfold was run using default settings with automated definition of the tissue segmentation. For histological mapping in the macaque BigMac brain, we manually corrected the tissue segmentation following nonlinear initialisation to fit the individual's anatomy optimally and then ran hippunfold in the BigMac brain as described above.

## Subfield drawing and microstructural mapping

Hippocampal subfields were manually labelled in QuPath v.0.2.3[107] onto the Cresyl Violet stained histological slices at a resolution of 0.28 μm/pixel (see Fig. 1D for an example and Supporting Data on Figshare for all slices). All histology data with annotations are provided open access (see Data Availability section). The cutting angle of the slices relative to the hippocampal long axis was approximately 50°, and the slice gap was approximately 0.35 mm (Fig. S1C). In the middle of the hippocampus, a section of approximately 2 mm was not covered by histological slices.

Subfield delineations were mainly based on previously described criteria in the macaque[114], which mirror those used in humans and previously for BigBrain[105]. We labelled CA3 and CA4 together as one subfield because the differentiation between the two was not consistently recognisable. One shared label was drawn for the subdivisions of the subiculum, referred to as the subicular complex. CA2 was evident by the high density of darkly stained neurons. The boundary

between CA1 and the subicular complex was determined by a drop in intensities corresponding to a change in the density of the pyramidal cell layer[76,115]. The alveus layer was not included in the subfield labels. Subfield boundaries were drawn roughly orthogonal to the intrinsic spiral axis rather than oblique to match the protocol previously used for the definition of human subfields in BigBrain[105]. Even though an oblique border is more commonly seen in anatomical literature[76], we adopted this simplification as it ensured a robust mapping to the hippocampal surface. Subfield annotations were exported to geojson format to apply for the TIRL registrations.

We then applied nonlinear slice-to-volume registrations to map the staining intensities from the Gallyas slices, the Cresyl Violet slices and the categorical subfield labels to the same individual's volumetric MGE space at a resolution of 0.15 mm³. For computational efficiency, only a hippocampal block was reconstructed in each hemisphere, rather than the whole brain. The volumetric microscopy data were then mapped to the high-resolution mid-thickness hippocampal surface. For representation in the flatmap space, the data were resampled to a regular matrix. To account for the gap between histological slices of the same contract, we performed linear interpolation (nearest-neighbour for subfield labels) in 2D space, which is topographically more suitable than volumetric interpolation, and smoothed the data (sigma = 3 mm). Data was sampled across all hippocampal surfaces and then averaged. The same workflow was applied to MRI metrics of microstructure following linear registration to the structural MGE volume using FSL's FLIRT v.6.0.

To account for slice-to-volume registration error and deviations in the labelling, the surface subfields map was manually corrected in GIMP v.2.8.22, guided by overlays of the microstructural surface maps as previously recommended[116]. This procedure ensured spatial continuity and plausibility of the map especially in the tail and head region of the hippocampus, where non-optimal cutting posed limitations on a serial manual labelling approach[117]. The surface mapping of the non-corrected raw subfield labels is shown in Fig. S1D. Note that the human BigBrain subfield labels were drawn in a dense 3D volumetric reconstruction of the histological data, which meant that spatial continuity was ensured, and no such correction was needed. Applying a 3D labelling approach, however, was not applicable to the BigMac dataset, given the sparser sampling of slices and, therefore, the larger slice gap.

## Subfield map quantification
We focussed quantifications of the subfield maps on the extent of the subfields along the distal-proximal axis, given the overall pattern of vertical stripes. First, we measured the global similarity of the subfield maps across hemispheres and species. Quantification of overlap (such as Dice) is not suitable for such a multilabel segmentation problem as an expansion in one subfield will affect the location of all other subfields. Therefore, we quantified the pairwise cosine distance of each row in the subfield map in a 4-dimensional space characterising the extent of the four subfields. The mean distance was then computed for each pair of subfield maps. Next, we quantified the relative size of each subfield compared with each other subfield. To quantify species differences, we computed the percentage change of this metric between humans and macaques.

## Macaque in vivo MRI pre-processing
In vivo, structural scans of the macaques were processed with an NHP adaptation of the openly available HCP pipeline[104]. We adapted the pipeline scripts to run with only T1w scans as T2w scans in the same individuals were not available. Further, we adapted the brain-extraction step with an inhouse-script from the MrCat toolbox (github.com/neuroecology/MrCat). Structural processing included, amongst others, the reconstruction of the individual's brain surface using FreeSurfer v.7.2[118] and registration to the Yerkes-19 space based on FSL's FLIRT[119] and FNIRT[120].

Volumetric processing of the resting-state fMRI scans was performed using a custom shell script pipeline of FSL commands (v.6.0[121]), as other automatic pipelines did not provide adequate results, particularly for brain extraction and cleaning. Initially, scans were reoriented and the five first volumes were discarded. Then the data were bias-corrected using FSL's FAST[122] and linearly registered with the structural scan using FLIRT. Motion correction using MCFLIRT and spatial smoothing (kernel FWHM = 2 mm) was performed using FSL's FEAT. The motion-corrected scans were further processed using ICA-based cleaning based on FSL's MELODIC. In a first step, the non-brain-extracted scans were processed using automatic estimation of dimensionality and manually classified noise components were removed. In a second step, the cleaned and brain-extracted scans were decomposed into 20 components, and the few remaining noise components were removed. The cleaned scans were then transformed to the group-level Yerkes19 template space based on non-linear registration. Following volumetric processing, the resting-state scans were processed with the fMRI surface processing part of the NHP–HCP pipeline. As part of the pipeline, the macaque fMRI data were mapped to each individual's brain surface in fsLR_10k space and converted into cifti-format, paralleling the available human fMRI data.

All following resting-state analyses were carried out in parallel for both species using the same tools and parameters unless otherwise specified. We use the conventional term cortex to refer to the HCP surface data, which covers mainly the neocortex. However, we do not mean to imply that the hippocampus is part of the subcortex. To assess the reliability of the fMRI-based species comparison, we derived voxel-wise temporal signal-to-noise-ratio (tSNR) images for each individual and averaged these for the group. We provide the volumetric and the hippocampal surface map of tSNR in the Supplemental Information (Fig. S1H).

## Resting-state data general processing
The volumetric part of the fMRI cifti-files was mapped to the low-resolution hippocampal surface in template space (MNI for humans and Yerkes19 for macaque) using the HCP's connectome workbench toolbox v.1.2.3 (wb_command[123], ribbon-constrained method). Resting-state data were then spatially smoothed on the cortical and the hippocampal surface (kernels for cortex/hippocampus: 6 mm/4 mm for the human and 2 mm/1 mm for the macaque). Prior to any further analyses, we regressed the mean time series out of each individual's resting-state data. To ease the computational burden for gradient analyses, cortical resting-state data of each hemisphere was parcellated into approximately 1000 parcels from an existing parcellation[124]. Note that the parcels have no correspondence across species and the parcellation only served the purpose of downsampling.

## Hippocampal gradients
The workflow for generating functional gradients was based on the Micapipe pipeline[125] (see Fig. 2A for a schematic). In each individual, we generated a connectivity matrix between parcellated cortical and hippocampal fMRI data based on time-series correlation, followed by Fisher R-to-Z transformation. The connectivity matrices for all individuals were averaged, followed by diffusion map embedding as implemented in BrainSpace v.0.1.4[126]. This step involved the generation of an affinity matrix using a normalised angle as an affinity metric. The embedding was performed for the left and right hippocampus separately, and each of them was embedded based on connectivity with cortical parcels from both hemispheres.

## Joint cross-species embedding
Next, we generated a joint cross-species cortico-hippocampal gradient (see Fig. 2C for a schematic). For each individual, we computed a connectivity matrix of all hippocampal vertices with all cortical parcels and averaged these for each species. Then, we concatenated the two

connectivity matrices for each species so that the cortical dimension was doubled in size and applied Fisher's transformation. Lastly, we generated an affinity matrix and applied diffusion map embedding as described above.

## Functional cortical gradients

Cortico-cortical gradients were derived based on cross-correlation of all cortical parcels in the left and right hemispheres, followed by Fisher R-to-Z transform, affinity kernel computation, and diffusion map embedding as described above (see Fig. 2D for a schematic). To test the relationship between cortico-cortical gradient maps and the 1st joint cross-species gradient map, we performed feature selection via LASSO regression as implemented in scikit-learn (alpha = 0.1). Prior to the regression, data were transformed to a Gaussian distribution using scikit-learn's QuantileTransformer. To compute a goodness of fit for each gradient map, we used Ordinary Least Squares regression as implemented in the Python statsmodels package. The significance of the correlations was determined using a spin test approach (1000 permutations) to control for spatial auto-correlations[127] as implemented in BrainSpace[126]. To quantify the overlap of gradient maps, we derived the Dice coefficient of the joint cross-species gradient and each cortical gradient map after applying a quantile-based threshold across a range of thresholds (70–95%). In addition, we computed the dice coefficient of the joint cross-species gradient with each pair and each triplet of cortical gradient maps.

## Cortico-hippocampal connectivity

The connectivity of the hippocampus with the cortex was determined using Pearson correlation between each pair of vertices (Supplemental Information, Fig. S3A). The maximal value across all hippocampal vertices was assigned to each cortical vertex to capture connectivity with any part of the hippocampus. Furthermore, we assessed whether the cortical connectivity pattern with different parts of the hippocampus is diverse (Supplemental Information, Fig. S3B). Therefore, we constructed four hippocampal ROIs, or sectors: anterior-medial, anterior-lateral, posterior-medial, and posterior-lateral hippocampus. The four sectors were defined based on their coordinates in the hippocampal flatmap space to ensure that geometrically matched ROIs were utilised in both species. Connectivity with four hippocampal sectors was derived based on correlation with the mean time series of each sector.

## Homology index

Finally, we obtained a whole-brain map of species homology or divergence[49,50], based on hippocampal connectivity (Supplemental Information, Fig. S3C). Similar to a previous description[49,50], we first applied a cross-species registration to establish the rough correspondence of cortical parcels across species. We accessed a previously developed surface registration, which was based on cortical myelin content[101]. Then, we derived a measure of homology for each human cortical parcel: We computed the median correlation with all macaque parcels within a searchlight (radius: 15 cm) based on the spatial connectivity profile with the hippocampus. For each of the seven human cortical networks, as defined based on the Yeo parcellation[51], we finally computed the mean correlation.

## Dual regression of hippocampal axes

To study the cortical reflection of the two hippocampal axes, we used a dual-regression approach[52,53]. For each individual, cortical and hippocampal resting-state data were concatenated in space to form a target 4D dataset. For each of the two hippocampal axes, we generated a regressor by stratifying the hippocampal flatmap (16 bins for the anterior-posterior axis and 8 bins for the distal-proximal axis) and assigning values ranging from −1 (most posterior or most distal) to 1 (most anterior or most proximal) in each bin. Those elements in the regressor corresponding to cortical and not hippocampal elements

were set to 0. The two hippocampal regressors and an additional regressor modelling an intercept formed an orthogonal design matrix. In the first stage of the dual-regression, we multiplied the pseudo-inverse of the design matrix with the data matrix, to obtain a single time-series for each hippocampal axis. In the second stage of the dual regression, we regressed this set of time series back into the data matrix. This operation results in spatial brain maps quantifying the functional connectivity with each of the regressors. All individual subject's brain maps were averaged.

## Reprojection from cortex to hippocampal space

As alternative visualisation, we used the two group-level spatial brain maps of the dual regression analysis as coordinates for a 2D space (see Fig. 3B for a schematic). This space reflects the intrinsic coordinates of the hippocampus itself and the aspect-ratio was adapted accordingly. The coordinates were linearly rescaled after excluding the top 10% of vertices on either end of the two axes. First, we selected a set of homologous vertices on the left hemisphere brain surface of an example subject in wb_view (Supplemental Information, Fig. S3D) and mapped these into the hippocampal space. The cross-species difference for each region in the 2D space was assessed using the Kolmogorov–Smirnov test with a significance threshold of $p < 0.01$ corrected for the number of comparisons performed. Next, we mapped all brain vertices of the joint cross-species gradient based on hippocampal connectivity embedding (the map shown in Fig. 2C) back into this space. The two hemispheres were combined for this visualisation.

## Reporting summary

Further information on research design is available in the Nature Portfolio Reporting Summary linked to this article.

## Data availability

High resolution Nissl-stained histology data with digital annotation files, the CIVM template MRI tissue segmentation, as well as ex vivo macaque MRI data have been deposited in the WIN's Digital Brain Bank platform[106] (dataset title: Hipmac project). Macaque in vivo MRI data have been made available via the Open Science Framework (https://osf.io/hke98/, DOI: 10.17605/OSF.IO/HKE98). Supporting Data with screenshots of the hippocampal sections with annotations are openly available on Figshare. HCP data are publicly available at https://www.humanconnectome.org/. The CIVM macaque post-mortem template is publicly available at https://civmvoxport.vm.duke.edu/ [34]. The BigMac data are available via the Digital Brain Bank platform[106] (dataset title: The BigMac dataset). The human BigBrain hippocampal subfield map was accessed from a previous study (https://zenodo.org/record/6360647 [105]).

## Code availability

Hippunfold is openly available as a BIDS App at https://github.com/khanlab/hippunfold. All code generated for this project to handle the cited software tools has been deposited at the Wellcome Centre of Integrative Neuroimaging's GitLab server[128]. The HCP NHP-Pipeline is openly available at https://github.com/Washington-University/NHPPipelines.

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

## Acknowledgements

N.E. is supported by a Sir Henry Wellcome Postdoctoral Fellowship from the Wellcome Trust [222799/Z/21/Z]. Jd.K. is supported by a Natural Sciences and Engineering Research Council of Canada—Postdoctoral Fellowship (NSERC-PDF). AFDH and INH were funded by the Wellcome Trust [WT215573/Z/19/Z]. KLM and the BigMac dataset were funded by the Wellcome Trust [WT202788/Z/16/Z]. S.Z. was supported by the Chinese Government Scholarship. S.J. and K.L.M. are supported by a Wellcome Collaborative Award [215573/Z/19/Z] and a Wellcome Senior Research Fellowship [221933/Z/20/Z]. B.C.B. acknowledges research support from the National Science and Engineering Research Council of Canada (NSERC Discovery-1304413), CIHR (FDN-154298, PJT-174995), SickKids Foundation (NI17-039), Helmholtz International BigBrain Analytics and Learning Laboratory (HIBALL), Healthy Brains and Healthy Lives (HBHL), BrainCanada, and the Tier-2 Canada Research Chairs programme. The Wellcome Centre for Integrative Neuroimaging is supported by core funding from the Wellcome Trust [203139/Z/16/Z and 203139/A/16/Z]. For the purpose of Open Access, the author has applied a CC BY public copyright licence to any Author Accepted Manuscript version arising from this submission.

## Author contributions

Conceptualisation: N.E. and B.C.B.; Methodology: N.E., J.D.K., A.F.D.H., I.N.H., S.Z., R.B.M., S.J., and B.C.B.; Software: J.D.K., I.N.H.; Resources: J.S., A.F.D.H., K.L.M., and R.B.M.; Writing—original draft: N.E. and B.C.B.; Writing—review & editing: all authors.

## Competing interests

The authors declare no competing interests.
