## [Peer Review File · Nature Communications]

Hippocampal connectivity patterns echo macroscale cortical evolution in the primate brainREVIEWER COMMENTS

Reviewer #1 (Remarks to the Author):

The authors employ novel methodology to compare hippocampal structure and connectivity between a human and a macaque. From the viewpoint of evolutionary biologists, there is considerable merit in so mapping the connectivity of the different axes of the hippocampus in a non-invasive manner to understand differences among species, in particular differences between model species and humans. Support for previous findings regarding an incomplete DMN and functional differentiation along the long axis in macaques are particularly interesting results. To fully ascertain the extent to which human hippocampal embedding and function is unique, however, would, as a start, require applying the authors' methods across many other species, in particular great apes. When attempting to interpret the evolution of the human hippocampus (especially when comparing it to only one model species), this significant limitation needs to be acknowledged, and the conclusions tampered. In addition, what the authors fail to clearly communicate are 1) their hypotheses/predictions and 2) why a general Nature Communications audience should be interested in these differences. Moreover, the manuscript is currently not up to publication standards in terms of wording/clarity, including numerous grammatical errors which still need to be addressed.

Reviewer #2 (Remarks to the Author):

I'd first like to congratulate the authors for their fine and detailed work exploring species differences in hippocampal structure and function. It is exciting to see functional imaging experiments with such a focus on anatomical detail. The paper does a very good job at underscoring the importance of characterising the structural and functional organisation of the hippocampus and how this may differ across species.

My suggestions below, rather than criticisms, largely reflect potential issues/questions that I believe, if addressed, may improve the clarity and interpretation of results.

On page 4; line 99-100 the authors state ". . . from distal to proximal (i.e., from the dentate gyrus to Cornu Ammonis, CA, and subiculum subfields)." However, the description in brackets seems not to match the distal-proximal, rather this seems to describe progression from proximal (closer to DG) to distal (further from DG)? If distal to proximal, should this not read, "i.e., from the subiculum subfields to Cornu Ammonis, CA and dentate gyrus"?

I kept asking myself, I wonder how did hippunfold perform in relation to subfields within the uncal region of the hippocampus which is smaller in the macaque than in the human? It might be useful to address this point in the methods section? Specifically, does hippunfold (which has been developed on the human brain), do a good job delineating subregions in the macaque uncus?

On a related note, Figure 1E displays the unfolded subfields but does not show the outer boundaries of the subfields. It would be informative to see the full extent of subfields in these figures, rather than have the edges cropped. Are there visualisable between species differences, particularly in the uncal and tail regions which are notoriously tricky areas to reliably segment? These regions cannot be seen in the current figure. If there are, could the authors speculate if, in their opinion, these differences relate to species differences or limitations in applying hippunfold to macaque data?

On pages 4-5 lines 116 – 118, the authors state that "We compared the macaque hippocampal map to that from the human BigBrain and the overall pattern was highly similar (similarity metric of 0.95 and 0.93 for left and right hemisphere". Does this refer to similarity for the whole hippocampus? Although the authors note that there are subtle differences in relative extent of subfields (i.e., CA2 and CA3/4) is it possible to calculate and report similarities for each subfield individually?

On page 7 lines 168-173, the authors report analyses to investigate whether macaque DMN comprises two distinct cortical networks and found support for this. Although not explicitly reported, it would be interesting to hear whether, in their opinion, to what degree did they find (or not find) evidence of multiple networks in the human, considering gathering evidence that the human DMN may also comprise separable networks (see Lee et al., 2021. *Journal of Neuroscience*, 41 (24) 5243-5250 for an example).

The authors display results relating to the 2nd component in human and the 6th component in macaque showing functional differentiation along the proximal-distal axis in supplemental figure 2A. Would it be possible to overlay schematic representations of subfield maps (similar to those displayed in Figure 1E) to better visualise how these gradients align with classically defined subfield locations? I believe this observation would be of interest to many in the hippocampal functional imaging community.

Related to this, the proximal-distal gradients displayed in Supplemental Figure 2A are not consistent along their anterior posterior axis in both species. Considering recent work showing differences in functional connectivity along the anterior-posterior axis of hippocampal subfields (for example see Dalton et al., 2019. *Neuroimage*, 192:38-51), I feel that this is an important observation that is a bit hidden in the current manuscript. This observation would be of great interest to the hippocampal subfields community and I feel the manuscript would benefit if it is noted in text, even if not discussed in depth.

At the end of the results section, the authors state that "Taken together with the histological results above we showed that the 'short' proximal-distal hippocampal axis captures microstructural variations of the hippocampus". It was unclear if this was in relation only to structure or to the observed proximal-distal functional gradient? If the latter, I'm unsure this claim can be made without further discussion of how these functional gradients align with microstructurally defined subfield boundaries?

On page 10, line 2, the authors state ". . . also to higher-order visual areas and ventral premotor cortex are mediated by the distal part of the hippocampus". To assist readers who may be less familiar with 'proximal-distal' terminology, could the authors clarify, in text, if 'distal' here refers to subicular regions?

MINOR

page 16 line 384, the authors state "these rapid expansions free up portions of cortex" should this be 'portions of cortex'?

Figure labels for Supplemental Figure S2 and S3 may be mistakenly labelled?

Reviewer #3 (Remarks to the Author):

The authors adapted Hipunfold, a tool recently developed to unfold the human hippocampus, to also create a surface-based coordinate system for the macaque hippocampus. This enabled them to perform a geometrically matched comparative analysis of the structural and functional organization of the human and macaque monkey hippocampi.

This is an interesting topic and timely study. The authors are expert computational neuroscientists with particular experience in the development of frameworks enabling comparative analyses. The results presented in this study provide further evidence supporting the preservation of hippocampal macro- and microstructure in primate evolution. Importantly, they demonstrate a marked evolutionary functional reconfiguration along the anterior-posterior hippocampal axis.

My main concern regards the correctness of the manual labels of hippocampal subfields. The authors state that "The following labels were manually segmented in ITK-SNAP v.3.8.0102: The hippocampal grey matter, the dentate gyrus, stratum radiatum, lacunosum and moleculare (SLRM, the 'hippocampal dark band'), the grey matter of the temporal lobe adjacent to the hippocampus,

the uncus, the hippocampal-amygdalar transition area, and indusium griseum (Supplemental Information, Figure S1A)”

Does this list of labels mean that the authors consider that the SLRM is not part of the hippocampal grey matter? If so, this is an anatomically incorrect assumption and represents a major flaw which confounds the results of the study as well as their interpretation.

Since Cajal’s first detailed description the hippocampal laminar structure in the 19th century there have been multiple reports providing a comprehensive description of the hippocampal laminar structure (e.g., Insausti & Amaral 2007; Palomero-Gallagher et al. 2022; Vida 2018; Witter 2018). In short: In the CA region the hippocampal grey matter consists of the stratum oriens (contains the basal dendrites of the pyramidal cells), stratum pyramidale (contains the cell bodies of the pyramidal cells), stratum radiatum (contains the proximal portion of the apical dendrites of the pyramidal cells), and stratum lacunosum-moleculare (contains the distal portion of the apical dendrites of the pyramidal cells). In the dentate gyrus the hippocampal grey matter encompasses the stratum moleculare (contains the apical dendrites of the granule cells), stratum granulosum (contains the cell bodies of the granule cells) and stratum multiforme. The alveus layer, which contains the axons of the pyramidal cells, is referred to as hippocampal white matter.

Figures 1D and S1A present manual labels which seem to have been performed following different criteria. It would be helpful for the reader if they included a legend. The delineation shown in figure 1A is overall correct except for the border between the CA1 region and the pre-/subiculum. The border between these two compartments runs oblique to the pial surface, as nicely highlighted by the myelin stain. Concerning manual segmentations in Figure S1: what is the structure labelled in yellow? Red and green could be interpreted as representing the pyramidal and radiatum/lacunosum-moleculare layers of the CA region, respectively. However, this can’t be so, because this would mean that the pyramidal layer would be overrepresented, particularly in the C2 and CA3 segments. Furthermore, both the red and green compartments extend into the subicular complex, thus overestimating the proximo-distal extent of CA. One blue compartment highlights part of the dentate gyrus. I guess that, analog to labels in Fig. 1D, the other blue compartment should represent the pre-/subiculum. But this is not the case. It also covers a large portion of the deeper layers of the entorhinal cortex.

To enable assessment of the correctness of anatomical labels used in the present study, authors should provide a (supplementary) figure demonstrating, for the entire rostral-caudal extent of the hippocampus, the manual segmentations which serve as the ground truth for their analysis.

We thank the Reviewers for their thorough critique and the constructive feedback, which greatly helped us to improve the manuscript. Below, we address in detail each of the points that were raised. Reviewer's comments were numbered and otherwise reproduced verbatim in black font and our response is provided in blue font. References to page numbers and figures refer to the clean version of the revised manuscript. A version with tracked changes is also provided.

Reviewer 1 (R1):

R1#2. The authors employ novel methodology to compare hippocampal structure and connectivity between a human and a macaque. From the viewpoint of evolutionary biologists, there is considerable merit in so mapping the connectivity of the different axes of the hippocampus in a non-invasive manner to understand differences among species, in particular differences between model species and humans. Support for previous findings regarding an incomplete DMN and functional differentiation along the long axis in macaques are particularly interesting results.

We thank the Reviewer for the positive feedback and have addressed their concerns in detail below.

R1#2. To fully ascertain the extent to which human hippocampal embedding and function is unique, however, would, as a start, require applying the authors' methods across many other species, in particular great apes.

We fully agree with the Reviewer's concern. Claiming the uniqueness of the human hippocampal embedding would require comparisons to other species, particularly those more closely related than macaque monkeys. We have therefore tempered our conclusions relating to this point (see response below, R1#3). However, we would like to explain why we cannot include the suggested analysis in our manuscript. Acquiring fMRI data in animals is typically considered an invasive procedure, which often requires anaesthesia or restraining the animals. Performing invasive research of this form on great apes is illegal in the UK, EU, USA and many other countries. Anecdotally, historical fMRI datasets from chimpanzees exist (acquired before 2009), but these cannot be used for our study due to significant quality differences, vastly differing anaesthesia protocols, and concerns from UK ethics and funding bodies, which govern our work.

In the present work, we carefully selected resources from macaques and humans so that a valid quantitative comparison is possible. Such high-quality imaging and histological data are increasingly becoming available for macaque monkeys due to recent efforts to standardise data collection and the construction of shared databases (Messinger et al. *Neuroimage*, 2021). Repeating our analysis in another species would require us to obtain densely sampled histology data together with high-resolution structural MRI data and fMRI data of comparable quality to that of the human. Such data are currently not available for us in another species, but we are in the process of establishing new collaborations that will allow us to do so in the future, for example in the marmoset monkey and the mouse. To nevertheless acknowledge the important concern of the Reviewer, and we have highlighted this limitation in the revised discussion and balanced some of the conclusions (see next comment, R1#3).

R1#3. When attempting to interpret the evolution of the human hippocampus (especially when comparing it to only one model species), this significant limitation needs to be acknowledged, and the conclusions tempered.

As the Reviewer correctly pointed out, our study can indicate species differences between humans and macaques, but cannot truly determine what is uniquely human with respect to all other species. Following the Reviewer's suggestion, we now discuss these limitations and have modified our conclusions:

The following paragraph was added to the discussion section:

Page 16: “Interpreting our findings in the wider context of evolutionary adaptations, ultimately requires an expansion of our framework towards more primate species and other mammals. Thanks to advances in the field of MRI, such a phylogenetic approach is becoming increasingly possible for whole-brain characteristics (Friedrich et al. *Neuroimage*, 2021), for example structural connectivity (Bryant et al. *PlosBiol* 2020; Assaf et al. *Nature Neuroscience*, 2020), or cortical folding (Heuer et al. *Cortex*, 2019), and brain function (Krubitzer et al. *Neuron*, 2007). Additionally, the growing integration of histology with MRI allows for cross-scale, cross-species investigations, as exemplified in the current paper. Promising advances in data sharing are also paving the way for functional connectivity fMRI studies (Gavin et al. *Cell reports*, 2022) to overcome the current limitation of small numbers of species.”

The following paragraphs were modified to tone down the conclusions:

Page 1 (abstract): “However, while functional organisation in both species also followed an anterior-posterior axis, the latter showed a marked reconfiguration across species [...].”

Page 13: “First of all, we found that the microstructural blueprint of the hippocampus, its principal functional axes in anterior-posterior direction, and cortical network embeddings are overall conserved across the two species. However, hippocampal embeddings to macroscale functional networks also reflected species differences, particularly in the default-mode network (DMN). Specifically, the inferior parietal lobe in the macaque mirrors an incomplete integration of the DMN compared to the human. Our findings, thus, suggest that the human DMN has expanded and further integrated in the human lineage, harnessing the hippocampal microcircuit in a potentially uniquely human way. Altogether these adaptations form the basis of specialised human brain function spanning a wide range of cognitive functions. Expanding our framework to a larger set of primate species will allow us to determine if these effects are truly unique to humans or gradually evolved across primates.”

R1#4. In addition, what the authors fail to clearly communicate are 1) their hypotheses/predictions and 2) why a general Nature Communications audience should be interested in these differences.

We now clarified our hypotheses and formulated a significance statement in the revised Introduction:

Page 4: “We provide a new comparative space that represents the hippocampus as an unfolded surface in which different species can be meaningfully compared. Notably, multimodal data (microscopy, structural and functional MRI) can be integrated within the same space, allowing simultaneous study of geometry, microstructure and function. We use this framework to study hippocampal differences between humans and macaques. We hypothesise that humans and macaques have highly similar hippocampal microstructure and that species differences relate to hippocampal connectivity with higher-order functional networks. Our methodology lays the groundwork for a new multi-modal comparative approach, where similar methods can be used to integrate information across other modalities and translate information across other species, to elucidate the evolutionary trajectories and underlying mechanisms shaping human cognitive abilities.”

R1#5. Moreover, the manuscript is currently not up to publication standards in terms of wording/clarity, including numerous grammatical errors which still need to be addressed.

We apologise for the oversight and have now identified and corrected the typographic and grammatical errors. For clarity of this response letter, we did not list all of these here, but instead refer the Reviewer to the attached revised manuscript with tracked changes.

Reviewer 2 (R2):

R2#1. I'd first like to congratulate the authors for their fine and detailed work exploring species differences in hippocampal structure and function. It is exciting to see functional imaging experiments with such a focus on anatomical detail. The paper does a very good job at underscoring the importance of characterising the structural and functional organisation of the hippocampus and how this may differ across species. My suggestions below, rather than criticisms, largely reflect potential issues/questions that I believe, if addressed, may improve the clarity and interpretation of results.

We thank the Reviewer for their positive evaluation and their detailed feedback, which we addressed below.

R2#2. On page 4; line 99-100 the authors state “. . . from distal to proximal (i.e., from the dentate gyrus to Cornu Ammonis, CA, and subiculum subfields).” However, the description in brackets seems not to match the distal-proximal, rather this seems to describe progression from proximal (closer to DG) to distal (further from DG)? If distal to proximal, should this not read, “i.e., from the subiculum subfields to Cornu Ammonis, CA and dentate gyrus”?

The terms ‘proximal’ and ‘distal’ here are used relative to the hippocampus, i.e. the most proximal structures are closer to the subiculum, whilst the more distal structures are further away from the subiculum, following Ding & Hoesen *JCompNeurol*, (2015). We prefer to stick with the current terminology, as it has previously been used in all hippunfolds-related publications and because these are the default output filenames when using the software (DeKraker et al. *eLife*, 2022). We have added a sentence to clarify the terminology to avoid any confusions:

Page 3: “Proximal in this context refers to the structures closer to the subiculum, whilst distal refers to those regions further away from the subiculum and closer to the dentate gyrus (Ding & Hoesen *JCompNeurol*, 2015)”.

R2#3. I kept asking myself, I wonder how did hippunfolds perform in relation to subfields within the uncus region of the hippocampus which is smaller in the macaque than in the human? It might be useful to address this point in the methods section? Specifically, does hippunfolds (which has been developed on the human brain), do a good job delineating subregions in the macaque uncus?

The Reviewer is raising an important point that we specifically paid attention to when designing the study. First of all, we would like to emphasise that the macaque hippocampal subfield definition was based on histology and was not predicted or generated by the hippunfolds tool. The same is true for the human subfields, which were manually defined on the BigBrain histological dataset in a previous study (DeKraker et al., *Neuroimage*, 2020). This is now stated more explicitly in the methods:

Page 19: “This human hippocampal subfield map was defined based on histology only.”

To aid the interpretation and visualisation of the macaque subfields maps, we now generated a new supplementary file (**Supplementary File 1**), which shows a 3D reconstruction of the macaque subfields together with the histological slices, a multiplanar view and an intersection with the unfolded 2D flatmap. The subfield definition in the uncus region is clearly visualised in the gif. We hope that this new visualisation will help the readers to appreciate the complex anatomy of the hippocampal subfields and their relation to the 2D flatmap representation. A screenshot of the gif is shown below:

Supplementary File 1. Hippocampal anatomy. Hippocampal subfields manually labelled in a reference macaque brain (BigMac, left hemisphere only). **Top left** - 3D rendering of the subfields. A coronal and an oblique plane, orthogonal to the long-axis, are shown as transparently in grey. **Top right** - The unfolded 2D subfield map with the intersection of the coronal plane and the y-coordinate measured from the back of the brain. **Bottom left/middle** - The intersection of the MRI volume with both planes. **Bottom right** - The histology slice approximately corresponding to the coronal section. The dentate gyrus (yellow) is included to provide orientation only.

Page 5: “An animated visualisation of the macaque hippocampal subfields is provided in **Supplementary File 1**, where we show a 3D rendering alongside the 2D flatmap and the histology slices, with the virtual cutting plane moving from posterior to anterior.”

Note that the hippunfold tool indeed has the capacity to predict subfields in a new brain. In this case, it can use topographic and geometric information to project a reference subfield map to a new brain. The validity of such predicted subfields in humans has been demonstrated in previous related papers at the level of human MRI scans (DeKraker et al. *eLife*, 2022), and we recently improved the algorithm to also perform a better alignment across individuals (DeKraker et al. *eLife*, 2023). Functionality of predicting subfields in macaque scans will soon be available in a newer version of hippunfold, as our current study defined and validated the reference map for the macaque brain.

R2#4. On a related note, Figure 1E displays the unfolded subfields but does not show the outer boundaries of the subfields. It would be informative to see the full extent of subfields in these figures, rather than have the edges cropped.

The flatmap actually displays the whole extent of the subfield-related parts of the hippocampus: For the short axis (*i.e.*, proximal-distal), ranging from the most proximal part of the subicular complex to CA3/4 bordering DG, and for the long axis (*i.e.*, anterior-posterior), ranging from the head of the hippocampus to the tail. Consequently, for projection to a rectangular space, the head and the tail are distorted and relatively expanded. The relative size of the surface vertex spacing and expansion map highlighting this effect are shown in **Supplementary Figure S1A**. As part of the topographic unfolding, the final tip of the tail and of the head are represented at the uppermost and lowermost edge of the rectangular map, respectively, whilst in volumetric space they are actually curled towards the body. This relationship is demonstrated in **Supplementary Figure S1B**:

Figure S1. Related to Figure 1. A: Low-resolution hippocampal flatmap mesh and surface area of each vertex. **B:** The volumetric order of histology slices from posterior to anterior mapped to the hippocampal flatmap. [...]

Based on long-established literature from histological investigation, we expect each subfield to be present along the whole long axis (Duvernoy, *The Human Hippocampus*, 2005) and this is what our results show (**Figure 1E** and **Supplemental Information, Figure S1D**). In the revised manuscript, we explicitly highlight these concepts now more clearly:

Page 4: “The flatmap covers all subfield-related parts of the hippocampus in full and the resulting surface expansion map is shown in **Supplemental Information, Figure S1A**. The tip of the hippocampal tail is represented at the uppermost edge of the flatmap. In volumetric space, however, it is located more anteriorly than the most posterior section of the hippocampus, due to the curling of the structure; similar for the head region as shown in **Supplemental Information, Figure S1B**”.

R2#5. Are there visualisable between species differences, particularly in the uncal and tail regions which are notoriously tricky areas to reliably segment? These regions cannot be seen in the current figure. If there are, could the authors speculate if, in their opinion, these differences relate to species differences or limitations in applying hippunfold to macaque data?

Please see our reply to the point above (R2#4) explaining that the uncal and tail region are shown in the flatmap and also in our new visualisation (**Supplementary File 1**). Delineating the subfields in serial histology sections of the uncal and tail regions is indeed very challenging, if not impossible. The reasons are, amongst others, that optimal cutting is impossible in a folded structure and that the slice gap exceeds the width of the archicortical layers (see **Figure S1D**). However, our framework offers a solution to this problem, which we hope will be of use to many anatomists: First, the hippocampal surface is reconstructed independently of the histology data, relying on a discrete tissue segmentation of anatomical structures. Secondly, hippocampal subfields were manually defined in serial 2D histology slices. The subfield labels were mapped to volumetric 3D space, using individual slice-to-volume registrations. The subfields were then sampled to the surface, using the intersection of the hippocampal midthickness surface with the volumetric version of the subfields. The resulting 2D ‘raw’ subfields map is shown in **Supplementary Figure S1F**. When drawing subfield labels in serial 2D histology slices, small inaccuracies are almost inevitable, and these are visible in the same figure. Instead of correcting

these laboriously in the 2D histology slices directly, we were now able to correct these in surface space, which is a topology-preserving representation of hippocampal anatomy. Importantly, we also sampled MRI metrics from the same individual to the same surface and these showed the expected intensity variations for each subfield (**Figure 1D**).

Finally, we think that the assignment of the 2D topographic coordinates by hippunfold is indeed introducing minor distortions towards the head of the hippocampus, visible as a twist of the subfields in the lower third of the surface map. However, we observed this effect in both species independently and therefore it is unlikely to have introduced or overshadowed any meaningful between-species differences. Please see below for the relative adjustments to the manuscript:

Page 5: “The unfolding algorithm introduced some mild distortions evident by a twist in the most anterior third of the subfield map (**Figure 1E**). This effect is independently observed in both species”.

R2#6. On pages 4-5 lines 116 – 118, the authors state that “We compared the macaque hippocampal map to that from the human BigBrain and the overall pattern was highly similar (similarity metric of 0.95 and 0.93 for left and right hemisphere”. Does this refer to similarity for the whole hippocampus? Although the authors note that there are subtle differences in relative extent of subfields (i.e., CA2 and CA3/4) is it possible to calculate and report similarities for each subfield individually?

This quantification is a summary metric for the hippocampus in one hemisphere. The differences in relative extent of subfields are reported in **Figure S1G**. The ‘& delta’ matrix panel shows the percentage difference in relative sizes of subfields between humans and macaques. In the revised version of the manuscript we now specifically refer to this result:

Page 5: “We compared the macaque hippocampal map to that from the human BigBrain and the overall pattern was highly similar (similarity metric of 0.95 and 0.93 for left and right hippocampus, across all subfields, **Supplemental Information, Figure S1F**). Subtle differences in relative extent of hippocampal subfields, however, as quantified using pairwise comparisons were also observed (**Supplemental Information, Figure S1G**). For example, CA2 and CA3/4 are relatively expanded in humans.”

R2#7. On page 7 lines 168-173, the authors report analyses to investigate whether macaque DMN comprises two distinct cortical networks and found support for this. Although not explicitly reported, it would be interesting to hear whether, in their opinion, to what degree did they find (or not find) evidence of multiple networks in the human, considering gathering evidence that the human DMN may also comprise separable networks (see Lee et al., 2021. *Journal of Neuroscience*, 41 (24) 5243-5250 for an example).

The Reviewer raises an interesting topic and we would like to comment on this point. The data decomposition method that we employed did not reveal a dissociation of the DMN in humans. Importantly, we used the same decomposition method in both species. Extended functional MRI scanning in humans has recently provided evidence for the presence of interdigitated subnetworks within heteromodal association systems such as the DMN (see e.g. Braga and Buckner, *Neuron*, 2017). Whether the two networks we observed here in macaques using different decomposition techniques relate to higher order network interdigitation remains to be investigated, ideally also using extended scanning paradigms in both humans and non-human primates, together with tailored approaches to study network idiosyncrasy across individuals (Benkarim et al. *Communications Biology*, 2021). Speculatively, we nevertheless believe that the more fractionated layout of the DMN in macaques relative to humans relates to a less pronounced anterior-posterior integration in the former species (Genon et al. *TrendsNeurosci*, 2021), as well as an incomplete functional evolution of the temporoparietal region in non-human primates compared to humans (Mars *PNAS*, 2013, Xu et al. *Neuroimage*, 2020).

R2#8. The authors display results relating to the 2nd component in human and the 6th component in macaque showing functional differentiation along the proximal-distal axis in supplemental figure 2A. Would it be possible to overlay schematic representations of subfield maps (similar to those displayed in Figure 1E) to better visualise how these gradients align with classically defined subfield locations? I believe this observation would be of interest to many in the hippocampal functional imaging community. We thank the Reviewer for the great suggestion and implemented it in the revised manuscript. **Supplemental Information, Figure S2A** is shown here with updated captions:

Figure S2. Related to Figure 2. A: Higher-order hippocampal gradients: 2nd human and 6th macaque hippocampal gradient displaying proximal-distal differentiation. The sign of the map is random. Subfield borders are indicated with black lines. [...]

R2#9. Related to this, the proximal-distal gradients displayed in Supplemental Figure 2A are not consistent along their anterior posterior axis in both species. Considering recent work showing differences in functional connectivity along the anterior-posterior axis of hippocampal subfields (for example see Dalton et al., 2019. *Neuroimage*, 192:38-51), I feel that this is an important observation that is a bit hidden in the current manuscript. This observation would be of great interest to the hippocampal subfields community and I feel the manuscript would benefit if it is noted in text, even if not discussed in depth.

We thank the Reviewer for bringing this aspect to our attention and following this suggestion, we now highlight this result more clearly in the main body of the revised manuscript. However, we refrained from an in-depth discussion of subfield-specific connectivity differences, given that the spatial resolution of the fMRI data is likely to introduce partial volume effects when considering individual subfields. Whilst the overall spatial trends across the spatial axes of the hippocampus are reliable, subfield-specific functional imaging with whole-brain coverage is currently still out of reach, especially for the macaque acquisition. In the paper that the Reviewer highlighted, for example, the authors acquired only a partial volume covering the temporal lobe and not the whole brain.

Page 7:” These proximal-distal gradients show variations that appear to differ across hippocampal subfields. This finding is consistent with recent reports on functional connectivity differences along the anterior-posterior axis of hippocampal subfields obtained using high-resolution precision scanning protocols (Dalton et al. *NeuroImage*, 2019). However, due to limits in spatial resolution of the fMRI data available for our study, these variations were not further investigated.”

R2#10. At the end of the results section, the authors state that “Taken together with the histological results above we showed that the ‘short’ proximal-distal hippocampal axis captures microstructural variations of the hippocampus”. It was unclear if this was in relation only to structure or to the observed proximal-distal functional gradient? If the latter, I’m unsure this claim can be made without further discussion of how these functional gradients align with microstructurally defined subfield boundaries?

We apologise for the unclear expression. This statement refers only to the structural/histological data and to the observation that all microstructural indices show the largest variation along the short axis and remain relatively constant along the long axis. We've adapted the sentence for clarity.

Page 9: "Taken together, we showed that the 'short' hippocampal axis primarily captures variations in microstructure, whilst the 'long' axis primarily characterises variations in functional organisation."

R2#11. On page 10, line 2, the authors state "... also to higher-order visual areas and ventral premotor cortex are mediated by the distal part of the hippocampus". To assist readers who may be less familiar with 'proximal-distal' terminology, could the authors clarify, in text, if 'distal' here refers to subicular regions?

We have adapted this section accordingly to facilitate the understanding of this result as we accidentally used the wrong term. As noted in our response further above (R2#2), we also clarify our terminology in the revised manuscript more clearly.

Page 11: "Connectivity to DMN nodes, but also to higher-order visual areas and ventral premotor cortex are mediated by more proximal, i.e. subicular, parts of the hippocampus."

R2#12. MINOR

page 16 line 384, the authors state "these rapid expansions free up portions of cortex" should this be 'portions of cortex'? Figure labels for Supplemental Figure S2 and S3 may be mistakenly labelled?

We apologise for these oversights and we have taken care to correct typos, and labelling mistakes in the updated manuscript. For clarity of this response letter, we do not list all of them here. In the revised manuscript, we now also consistently display the left hippocampus in **Figure 1**.

Page 6: **Figure 1** with minor adjustments regarding the display (the caption was not changed).

Reviewer 3 (R3):

R3#1. The authors adapted Hipunfold, a tool recently developed to unfold the human hippocampus, to also create a surface-based coordinate system for the macaque hippocampus. This enabled them to perform a geometrically matched comparative analysis of the structural and functional organization of the human and macaque monkey hippocampi.

This is an interesting topic and timely study. The authors are expert computational neuroscientists with particular experience in the development of frameworks enabling comparative analyses. The results presented in this study provide further evidence supporting the preservation of hippocampal macro- and microstructure in primate evolution. Importantly, they demonstrate a marked evolutionary functional reconfiguration along the anterior-posterior hippocampal axis.

We are very happy to read the Reviewer's overall positive assessment and appreciate their detailed feedback, which helped us to improve the anatomical accuracy of our study.

R3#2. My main concern regards the correctness of the manual labels of hippocampal subfields. The authors state that "The following labels were manually segmented in ITK-SNAP v.3.8.0102: The hippocampal grey matter, the dentate gyrus, stratum radiatum, lacunosum and moleculare (SLRM, the 'hippocampal dark band'), the grey matter of the temporal lobe adjacent to the hippocampus, the uncus, the hippocampal-amygdalar transition area, and indusium griseum (Supplemental Information, Figure S1A)". Does this list of labels mean that the authors consider that the SLRM is not part of the hippocampal grey matter? If so, this is an anatomically incorrect assumption and represents a major flaw which confounds the results of the study as well as their interpretation.

We thank the Reviewer for raising this point and we'd like to clarify here. The SLRM is drawn within the MRI volume as a relatively 'thin' sheet with roughly constant thickness throughout the hippocampus. It essentially serves only as the definition of the 'cutting plane' to unfold the hippocampus. Therefore, the definition of the SLRM does not represent a confound for our study, but it is a necessary landmark for the unfolding. Importantly, the SLRM is not a structure that we labelled within the histology slices and the MRI tissue segmentation that the Reviewer refers to is different from the histological subfield labelling, as clarified in more detail in our response to the question below (R3#4).

To address the Reviewer's comment and for clarification, we have now adapted the manuscript accordingly refraining from the use of the term 'grey matter' when referring to the subfield-related parts of the hippocampus, as in this example:

Page 21: "[...] The subfield-related regions of the hippocampus (Cornu Ammonis, CA), the dentate gyrus (DG), stratum radiatum, lacunosum and moleculare (SLRM, the 'hippocampal dark band'), the grey matter of the temporal lobe directly adjacent to the hippocampus, [...]"

The SRLM is typically the only intra-hippocampal feature visible in standard MRI sequences, hence the term 'dark band' is an intuitive and recognizable description (as detailed in Yushkevich et al. *Neuroimage*, 2015). As we noted in a previous hippunfolds publication (DeKraker et al. *Neuroimage*, 2018), the contrast of this structure may not only be driven by cell bodies and axons, but also by blood vessels and residual cerebrospinal fluid lining the hippocampal sulcus. These stipulations are noted in the referenced methods as they were previously established in our human work. We have added a clarification to the manuscript to avoid confusions:

Page 21: "The SLRM label covers a heterogenous set of structures including the archicortical cortex, but also axons and other incidental structures, such as residual cerebrospinal fluid. "

R3#3. Since Cajal's first detailed description of the hippocampal laminar structure in the 19th century there have been multiple reports providing a comprehensive description of the hippocampal laminar structure (e.g., Insausti & Amaral 2007; Palomero-Gallagher et al. 2022; Vida 2018; Witter 2018). In short: In the CA region the hippocampal grey matter consists of the stratum oriens (contains the basal dendrites of the pyramidal cells), stratum pyramidale (contains the cell bodies of the pyramidal cells), stratum radiatum (contains the proximal portion of the apical dendrites of the pyramidal cells), and stratum lacunosum-moleculare (contains the distal portion of the apical dendrites of the pyramidal cells). In the dentate gyrus the hippocampal grey matter encompasses the stratum moleculare (contains the apical dendrites of the granule cells), stratum granulosum (contains the cell bodies of the granule cells) and stratum multiforme. The alveus layer, which contains the axons of the pyramidal cells, is referred to as hippocampal white matter.

We thank the Reviewer for raising this point and we have carefully revised all subfield labels to exclude the alveus layer. Note that this change does not affect any of the presented results as the flatmap is derived from an intersection with the midthickness hippocampal surface. We added several clarifications in the text:

Page 23: "The volumetric microscopy data were then mapped to the high-resolution midthickness hippocampal surface."

Page 23: "The boundary between CA1 and pre-/subiculum was determined by a drop in intensity corresponding to a change in density within the pyramidal cell layer (Insausti & Amaral *JCompNeurol* 2007; Palomero-Gallagher et al. *BSF*, 2020). The alveus layer was not included in the subfield labels."

R3#4. Figures 1D and S1A present manual labels which seem to have been performed following different criteria. It would be helpful for the reader if they included a legend.

The Reviewer is correct that these two figures represent two separate segmentations used for different purposes. **Figure 1D** shows the hippocampal subfield labels, which were based on histological assessment in the BigMac resource. **Previous Figure S1A**, however, represents the tissue segmentation of anatomical structures, which were drawn in a template MRI volume and these serve merely to guide the unfolding algorithm. Most of the labels in the MRI-based tissue segmentation are of no interest anatomically, they are simply required for the hippunfold algorithm to separate the other labels and determine the overall orientation of the brain. They are only used for the unfolding but nothing more. We realise that this figure can be confusing for a reader who is not familiar with hippunfold as it looks similar to the labelled subfields. Therefore, we decided to remove the figure from the revised manuscript, but to keep the description in the methods section for reproducibility. Furthermore, we modified the manuscript to use the terms 'tissue segmentation' and 'subfield labels' consistently to avoid confusions and added a sentence for clarification:

Page 21: "Note, that this tissue segmentation should not be confused with the definition of hippocampal subfield labels, which is based on histology, as described below."

R3#5. The delineation shown in figure 1A is overall correct except for the border between the CA1 region and the pre-/subiculum. The border between these two compartments runs oblique to the pial surface, as nicely highlighted by the myelin stain.

The 2D subfields flatmap (**Figure 1E**) is derived by intersecting the subfield labels with the midthickness surface of the hippocampal grey matter. For this reason, defining an oblique border will effectively change neither the results nor our conclusions. For the same reason, the exact outer border of stratum oriens and the inner border of stratum lacunosum-moleculare does not affect our quantifications. The notion that the CA1-subicular complex border should be oblique is commonly seen in histology work (for example Palomero-Gallagher *BSF*, 2020), but is not universal. Importantly, our study capitalised on establishing a comparison between the two species. Therefore, we closely mimicked the subfield

labelling protocol that was originally used for the human BigBrain (DeKraker et al. *Neuroimage*, 2020), which had adopted a non-oblique border definition following, for example Ding & Hoesen, *JCompNeurol* (2015). In the revised manuscript, we've included an explicit statement for clarification:

Page 23: "Subfield boundaries were drawn roughly orthogonal to the intrinsic spiral axis rather than oblique to match the protocol previously used for the definition of human subfields in BigBrain (DeKraker et al. *Neuroimage*, 2020). Even though an oblique border is more commonly seen in anatomical literature (Palomero-Gallagher *BSF*, 2020), we adopted this simplification as it ensured a robust mapping to the hippocampal surface."

R3#6. Concerning manual segmentations in Figure S1: what is the structure labelled in yellow? Red and green could be interpreted as representing the pyramidal and radiatum/lacunosum-molecular layers of the CA region, respectively. However, this can't be so, because this would mean that the pyramidal layer would be overrepresented, particularly in the C2 and CA3 segments. Furthermore, both the red and green compartments extend into the subicular complex, thus overestimating the proximo-distal extent of CA. One blue compartment highlights part of the dentate gyrus. I guess that, analog to labels in Fig. 1D, the other blue compartment should represent the pre-/subiculum. But this is not the case. It also covers a large portion of the deeper layers of the entorhinal cortex.

As described above, we would like to apologise for the lack of clarity in this figure as these structures do not represent individual subfield labels, nor archicortical layers. The extension of the red and green compartment into the subicular complex is intended, as the red structure includes hippocampal subfield-related components, including the subicular complex. As noted in the referenced Method papers, this does exclude part of the parasubiculum due to constraints of the MRI contrast, which necessitates some heuristic simplification. As described above, **Previous Figure S1A** has now been removed from the manuscript to avoid confusion with the subfield annotations of this study.

R3#7. To enable assessment of the correctness of anatomical labels used in the present study, authors should provide a (supplementary) figure demonstrating, for the entire rostro-caudal extent of the hippocampus, the manual segmentations which serve as the ground truth for their analysis.

We fully agree with the Reviewer that such an additional visualisation will greatly improve the manuscript. We now generated two new Supplemental Figures: In our new animated file (**Supplemental Information, File 1**), we show a 3D reconstruction of the macaque subfields together with an oblique and a coronal plane and intersections with the unfolded 2D flatmap. Additionally, we show the approximately corresponding histological slice. The figure is animated and consecutively 'moves through' the entire long hippocampal axis. A screenshot of the animation is shown below. The manuscript has been edited as follows:

Page 5: "An animated visualisation of the macaque hippocampal subfields is provided in **Supplemental Information, File 1**, where we show a 3D rendering alongside the 2D flatmap and the histology slices, with the virtual cutting plane moving from posterior to anterior."

In addition we have generated the new **Supplemental Information, Figure S4** which shows more example histology slices. All digital annotation files associated with each of the labelled 83 histology slices will be made openly available for re-use by others upon publication. The exact subfield border definition cannot be considered the 'ground truth' basis for our functional analysis, as we did not perform any subfield-specific functional connectivity analyses. The new figure is noted in the revised manuscript:

Page 22: "Hippocampal subfields were manually labelled in QuPath v.0.2.3 onto the Cresyl Violet stained histological slices (see **Figure 1D** and **Supplemental Information, Figure S4** for examples)."

Supplementary File 1. Hippocampal anatomy. Hippocampal subfields manually labelled in a reference macaque brain (BigMac) are shown in several displays (left hemisphere only). **Top left** - 3D rendering of the hippocampal subfields. A coronal and an oblique plane, orthogonal to the long-axis, are shown as transparent grey planes. **Top right** - The unfolded 2D flatmap of subfields shown with the intersection of the coronal plane and the y-coordinate measured from the back of the brain. **Bottom left/middle** - The intersection of the MRI volume and subfields with both planes. **Bottom right** - The histology slice approximately corresponding to the coronal section. The dentate gyrus (yellow) is included to provide orientation only.

Supplementary Figure 4. Hippocampal subfield labels. Example Cresyl Violet slices from BigMac with manual subfield labels overlaid as drawn in QuPath. The y-coordinate indicates the distance from the back of the brain.

REVIEWER COMMENTS

Reviewer #1 (Remarks to the Author):

As previously stated, the study makes an interesting and worthwhile contribution to the literature. The revisions have largely addressed the concerns of the reviewers, but not entirely. The toned-down conclusions regarding the generalizability of the findings from two species to all primates represent a significant improvement. However, in all the documents I have received, including those with tracked changes, I see few to no corrections to the numerous awkward phrasings and punctuation errors that characterize the manuscript. The first paragraph of the introduction is emblematic of this problem.

Reviewer #2 (Remarks to the Author):

I thank the authors for their thorough rebuttals/changes and I agree that they have satisfactorily addressed the reviewers comments/concerns with one exception.

In the authors response to R2#2, relating to proximal-distal terminology, they suggest adding the following sentence to clarify their definitions: "Proximal in this context refers to the structures closer to the subiculum, whilst distal refers to those regions further away from the subiculum and closer to the dentate gyrus (Ding & Hoesen JCompNeurol, 2015)". This stands in contrast to quite well-established terminological conventions relating to proximal-distal hippocampal structure. The standard convention is that 'proximal' refers to closer to DG and 'distal' refers to further from DG. In contrast to this established DG-centric convention, the authors adopt a 'subiculum-centric' definition.

As a rationale, they state in their response that "the most proximal structures are closer to the subiculum, whilst the more distal structures are further away from the subiculum, following Ding & Hoesen JCompNeurol, (2015)". I disagree that this is Ding and van Hoesen's stance in their 2015 paper but I believe this statement hints at the possibility that the authors have potentially misinterpreted a specific passage of the Ding & van Hoesen 2015 paper?

Specifically, on page 2242 of the Ding and van Hoesen 2015 paper, they use the terms 'proximal' and 'distal' in a discussion of different staining characteristics of the presubiculum (PrS). They clarify here that 'proximal' and 'distal' PrS refers to 'near' and 'away from' the subiculum respectively. However, here they refer specifically and only to the PrS, which lies medial to the subiculum. For structures medial to the subiculum, near and away from the subiculum is also near and away from the dentate gyrus and therefore, the terms proximal/distal align with the conventional definitions (i.e., proximal to distal = DG - CA3 - CA2 - CA1 - subiculum - presubiculum - parasubiculum). However, when taking a subiculum-centric definition, as the authors do, moving away from the subiculum in a lateral direction (i.e., towards CA1 - CA2 - CA3) will be moving from proximal to distal as the authors note in their authors response: ". . . distal refers to those regions further away from the subiculum and closer to the dentate gyrus". This is against the flow of the conventional definitions and this was not proposed in the Ding and van Hoesen 2015 paper as cited by the authors in their response.

To summarise, Ding and van Hoesen stated that in relation to the PrS, proximal/distal refers to parts of the PrS near/away from the subiculum in the medial direction, aligning with conventional terminology. I do not believe that they state or infer that this would be the case moving away from the subiculum in the lateral direction (i.e., closer to the DG). Is it possible that the authors made a mistaken inference relating to this? For a final emphasis of Ding & Van Hoesen's use of the conventional terminology, they later refer to the 'distal' subiculum in reference to the part of the subiculum further away from the DG (i.e., the medial part) on page 2248.

By pressing this point, I mean no criticism. You have done such a beautiful job of bridging the divide between histological and neuroimaging investigations of the hippocampus. Rather, while this

may seem a minor point, although you define your use of the terms proximal/distal in relation to the subiculum, your definition stands in such contrast to the current conventional uses that it risks paradoxically increasing potential confusion and miscommunication between the histology and neuroimaging communities.

I would normally take quite a firm stance here, however, I understand there is a broader issue for the authors in maintaining consistency with the terminology if they have already used it in previous hippocampal-related publications and default output filenames. This is difficult to reconcile but I think, as a bare minimum, it will be important for the authors to explicitly state how their proximal-distal definitions differ from those that are conventionally used in the literature, if not why.

Reviewer #3 (Remarks to the Author):

The reviewed version of the manuscript is improved as compared to the original version. However, I still have some major concerns, most of which pertain the fact that although the authors have anatomical knowledge, they are not expert neuroanatomists. This results in application of incorrect anatomical labels to the identified structures and subsequently in a flawed view of hippocampal connectivity because the results are interpreted under the assumption that a given anatomical structure served as the basis for the analysis. This represents the major confound of the study, and this aspect of the manuscript still needs considerable improvement.

In the rebuttal letter the authors write that: "The SLRM is drawn within the MRI volume as a relatively 'thin' sheet with roughly constant thickness throughout the hippocampus.". However, the thickness of the radiatum and lacunosum-molecular layers is not constant throughout the hippocampus. It is proportionately thickest in the CA3 region and thinnest in the CA2 region. Furthermore, what the authors have identified in the MRI volume is the so-called "dark band", which refers to a hypointense line appearing within the grey matter of the hippocampus on high resolution T2 images, and which does not correspond to the entire width of the SRLM. Therefore, throughout the manuscript the authors should refrain from using the term SRLM for their delineations in the MR sequence. Rather, they should simply describe what they have labelled, namely the "so-called dark band or hypointense line appearing within the grey matter of the hippocampus between the pyramidal layer of the CA region and the molecular layer of the dentate gyrus". This is particularly true for the Material and methods section.

The statement "(SLRM, the 'hippocampal dark band'" on page 21 must be changed to reflect the following facts: The SLRM is the outer molecular layer of the CA3 region and in MRI images a hypointense line is found roughly at this topographical location.

The statement „SLRM label covers a heterogenous set of structures including the archicortical cortex, but also axons and other incidental structures, such as residual cerebrospinal fluid.“ On page 21 must also be changed. „archicortical cortex“ is redundant wording. It is a well established fact that the cortical ribbon also contains axons. Therefore "also axons" is also redundant. The important fact is that the SLRM label covers *part of the hippocampal molecular layer* and also incidental structures, such as residual cerebrospinal fluid. This is precisely why it is not a good idea to call this label SLRM!

It is worrying that the authors consider that the labels needed to guide the unfolding algorithm of HipUnfold are of no interest anatomically, since (if I understood correctly) the tool was developed to provide the framework for the analysis of anatomically relevant scientific questions.

Supplementary file 1 is a nice figure for a presentation, but of little scientific use, as it does not enable assessment of the correctness of anatomical labels.

Supplementary figure 4 only shows four sections and the resolution of the images provided is not sufficient to enable assessment, e.g., of the correctness of the CA3 label. I can only repeat my request. The authors should provide a figure demonstrating, for the entire rostro-caudal extent of the hippocampus (and at an adequate resolution), their manual segmentations. Alternatively, as part of the review process the reviewers should be able to access the 83 histology slices that the authors plan to make available upon publication. Even if the parcellation is not relevant for the connectivity analysis, it must be controlled from the anatomical point of view

Minor points:

- The authors should label which part of Fig. 1E shows the macaque (BigMac) hippocampal subfields and which the human (BigBrain) subfields.
- In Fig. 1E the authors have labelled (among others) the CA1 and pre-/subiculum regions. The prosubiculum is found between the CA1 region and the Subiculum. Is this area encompassed by the CA1 label or by the pre-/subiculum label?
- What is the meaning of the abbreviation a.u. in Fig. 1C?

Response letter

We thank the Reviewers for their positive evaluation and the constructive comments, which we feel have significantly improved our manuscript. Please find detailed responses to all comments below in blue font. The corresponding changes were highlighted in the revised manuscript, which is provided as 'clean' version and as 'tracked-changes' version.

Additional data requested by Reviewer 3 are provided via an external repository that can be accessed via a private link at: [REDACTED] The

password to open the two zip folders is: [REDACTED] The 'Wiki-page' of the repository (visible when expanded with 'Read more') explains how the histology data and annotation files can be loaded into the QuPath image software. We please ask the Reviewers to keep this link and data private, and to use it only for the purpose of this Review. Once the paper is accepted, these data will be made openly available for the Readers via Oxford WIN's Digital Brainbank. An additional 'Supporting Data' file with screenshots of histological slices is provided as part of the submission via a private link at:

[REDACTED]

Reviewer #1 (Remarks to the Author):

R1#1. As previously stated, the study makes an interesting and worthwhile contribution to the literature. The revisions have largely addressed the concerns of the reviewers, but not entirely. The toned-down conclusions regarding the generalizability of the findings from two species to all primates represent a significant improvement.

We thank the Reviewer for the positive evaluation and for recognizing that our previous revisions represented a significant improvement to the paper.

R1#2. However, in all the documents I have received, including those with tracked changes, I see few to no corrections to the numerous awkward phrasings and punctuation errors that characterize the manuscript. The first paragraph of the introduction is emblematic of this problem.

As suggested, we have carefully re-read the paper and revised several passages for clarity and adequate spelling, including the first paragraph of the introduction. We also verified the punctuation. Several native speakers on the author line were also involved. Please find the revised first paragraph below.

Page 2: *"The hippocampus is one of the most extensively studied parts of the brain¹. It is implicated in numerous cognitive and affective processes, associated with multiple brain networks, and a model region to examine how neural structure and function covary in space²⁻⁴. The hippocampus is also markedly affected in multiple common and detrimental indications, including neurodegenerative disorders^{5,6}, drug-resistant epilepsy^{7,8}, as well as psychiatric conditions^{9,10}. The hippocampal grey matter consists of archicortex, a phylogenetically old type of cortex, which is considered conserved across mammals¹¹. This evolutionary conservation is the basis for translational cross-species frameworks, and we have gained a deep understanding for hippocampal anatomy and function from model species, such as non-human primates^{12,13}. Yet, it seems contradictory that the hippocampus supports many functions sometimes considered unique to humans, such as autobiographical memory¹⁴, future thinking¹⁵, and self-perception¹⁶. This apparent paradox can be resolved by two potential explanations: Firstly, it is possible that species differences in primate hippocampal structure have been overlooked as evolutionary diversification of hippocampal anatomy has rarely been studied (but see¹⁷). Therefore, we need quantitative frameworks that go beyond measuring regional brain volumes to compare species. Or, secondly, the integration of the hippocampus within the rest of the brain has undergone fundamental reconfiguration since the last common ancestor between humans and monkeys. Species-specific specialisations in subcortical structures such as the striatum^{18,19} or the amygdala²⁰ support the second hypothesis."*

Reviewer #2 (Remarks to the Author):

R2#1. I thank the authors for their thorough rebuttals/changes and I agree that they have satisfactorily addressed the reviewers comments/concerns with one exception.

We thank the Reviewer for the positive evaluation of our response and the thoughtful remaining suggestion.

R2#2. In the authors response to R2#2, relating to proximal-distal terminology, they suggest adding the following sentence to clarify their definitions: “Proximal in this context refers to the structures closer to the subiculum, whilst distal refers to those regions further away from the subiculum and closer to the dentate gyrus (Ding & Hoesen JCompNeurol, 2015)”. This stands in contrast to quite well-established terminological conventions relating to proximal-distal hippocampal structure. The standard convention is that 'proximal' refers to closer to DG and 'distal' refers to further from DG. In contrast to this established DG-centric convention, the authors adopt a 'subiculum-centric' definition.

As a rationale, they state in their response that “the most proximal structures are closer to the subiculum, whilst the more distal structures are further away from the subiculum, following Ding & Hoesen JCompNeurol, (2015)”. I disagree that this is Ding and van Hoesen’s stance in their 2015 paper but I believe this statement hints at the possibility that the authors have potentially misinterpreted a specific passage of the Ding & van Hoesen 2015 paper?

Specifically, on page 2242 of the Ding and van Hoesen 2015 paper, they use the terms ‘proximal’ and ‘distal’ in a discussion of different staining characteristics of the presubiculum (PrS). They clarify here that ‘proximal’ and ‘distal’ PrS refers to ‘near’ and ‘away from’ the subiculum respectively. However, here they refer specifically and only to the PrS, which lies medial to the subiculum. For structures medial to the subiculum, near and away from the subiculum is also near and away from the dentate gyrus and therefore, the terms proximal/distal align with the conventional definitions (i.e., proximal to distal = DG - CA3 - CA2 - CA1 - subiculum - presubiculum - parasubiculum). However, when taking a subiculum-centric definition, as the authors do, moving away from the subiculum in a lateral direction (i.e., towards CA1 - CA2 - CA3) will be moving from proximal to distal as the authors note in their authors response: “. . . distal refers to those regions further away from the subiculum and closer to the dentate gyrus”. This is against the flow of the conventional definitions and this was not proposed in the Ding and van Hoesen 2015 paper as cited by the authors in their response.

To summarise, Ding and van Hoesen stated that in relation to the PrS, proximal/distal refers to parts of the PrS near/away from the subiculum in the medial direction, aligning with conventional terminology. I do not believe that they state or infer that this would be the case moving away from the subiculum in the lateral direction (i.e., closer to the DG). Is it possible that the authors made a mistaken inference relating to this? For a final emphasis of Ding & Van Hoesen’s use of the conventional terminology, they later refer to the ‘distal’ subiculum in reference to the part of the subiculum further away from the DG (i.e., the medial part) on page 2248.

By pressing this point, I mean no criticism. You have done such a beautiful job of bridging the divide between histological and neuroimaging investigations of the hippocampus. Rather, while this may seem a minor point, although you define your use of the terms proximal/distal in relation to the subiculum, your definition stands in such contrast to the current conventional uses that it risks paradoxically increasing potential confusion and miscommunication between the histology and neuroimaging communities.

I would normally take quite a firm stance here, however, I understand there is a broader issue for the authors in maintaining consistency with the terminology if they have already used it in previous

hippunfold-related publications and default output filenames. This is difficult to reconcile but I think, as a bare minimum, it will be important for the authors to explicitly state how their proximal-distal definitions differ from those that are conventionally used in the literature, if not why.

We thank the Reviewer for the detailed explanation and guidance. As suggested, we revised the manuscript, adopting the terminology preferred by the Reviewer. For example, we adapted the use of the terms 'proximal' and 'distal' and changed the figures accordingly. For the updated definition, please see the changes to the manuscript below:

Page 4-5: "*The unfolded flatmap space is defined based on intrinsic coordinates of the hippocampus, ranging from posterior to anterior (from tail to body and head) and from distal to proximal. Proximal in this context refers to the structures closer to the dentate gyrus, whilst distal refers to those regions closer to the subiculum³⁵. Note, this DG-centric terminology differs from the terminology used in previous related hippunfold-publications, which used terms relative to the neocortex.*"

Reviewer #3 (Remarks to the Author):

R3#1. The reviewed version of the manuscript is improved as compared to the original version.

We thank the Reviewer for recognizing the improvements made to our work and for the suggestions, which we addressed below.

R3#2. However, I still have some major concerns, most of which pertain the fact that although the authors have anatomical knowledge, they are not expert neuroanatomists. This results in application of incorrect anatomical labels to the identified structures and subsequently in a flawed view of hippocampal connectivity because the results are interpreted under the assumption that a given anatomical structure served as the basis for the analysis. This represents the major confound of the study, and this aspect of the manuscript still needs considerable improvement.

We fully agree with the Reviewer that deriving manual anatomical segmentations is a complex challenge that requires both care and expertise, especially when it comes to the intricate anatomy of the hippocampus and surrounding structures. In this direct quote from the foundational white paper by the Hippocampal Subfield Consortium (HSC), some of the relevant challenges are outlined succinctly (Yuskevitch et al. *Neuroimage* 2015, Page 4):

"[...] the anatomy of the human MTL is complex and variable, and the boundaries between different subfields have been described in the neuroanatomy literature using cytoarchitectonic features that require histological staining and microscopic resolution to visualize (Lorente de Nó, 1934; Rosene and Van Hoesen, 1987; Gloor, 1997; Insausti and Amaral, 2004; Duvernoy, 2005; Amaral and Lavenex, 2007; van Strien et al., 2012). Even at that resolution, neuroanatomical references do not always agree on the definition and boundaries of subfields. Any protocol that attempts to label these substructures in MRI, regardless of resolution, has to employ some combination of image intensity cues, known anatomical landmarks, and geometrical rules to define boundaries between substructures. A substantial number of manual segmentation protocols have been published in the last few years, and up to now, no common set of rules has been adopted by the research community."

Our original human MRI tissue segmentation was developed in close collaboration with senior neuroanatomists and formulated based on 'BigBrain', a reference human histology resource (see for example, DeKraker et al. *elife* 2023, 2020). The segmentation and histological correspondence was critically assessed by members of the HSC and further validated using data-driven and observer-independent clustering approaches (DeKraker et al. *elife*, 2020). Also in the present study, we validated the surface space using observer-independent microstructural markers. The

neuroanatomical basis of each anatomical label and their limitations are discussed in depth in the referenced method papers. It is, unfortunately, out of scope for our manuscript to reiterate all relevant considerations that went into developing the criteria for these labels.

More generally speaking, there is considerable variability in anatomical criteria within and across raters, as well as protocols, both at the level of MRI and histology (Olsen et al. 2019, *JPND*, Wuestefeld 2024 *bioRxiv*). Our approach therefore focuses on identifying robust and reliable landmarks that are critical for the computational unfolding of the hippocampal cortex (hence the name hippunfold) which allows us to represent this cortical region as a 2D sheet, which then serves as a continuous reference frame. Importantly, analytic estimation of distal-proximal and anterior-posterior coordinates allows for the analysis of subregional correspondence in a cross-species setting, which is the main purpose of this manuscript. The utility and potential of an unfolded hippocampal surface-space has, in fact, already been recognized more than 20 years ago (Zeineh et al., *The Anatomical Record*, 2001), and surface-based approaches have furthermore been validated in clinical populations with histological information on hippocampal pathology (Bernhardt et al., *Annals of Neurology*, 2016).

The hippunfold unfolding paradigm is purposefully designed to work with MRI data and does itself not rely on histological data. MRI is intrinsically 3D and independent of the cutting plane, has firmly established neuroinformatics for continuous analysis, can be harnessed *in-vivo* and is highly scalable. This means our analysis framework is also accessible to researchers who do not have histological data available, or do not have the necessary training to work with such data.

For this comparative study, we chose to combine MRI with histology, as they have complementary strengths and weaknesses to study primate anatomy. By doing so, we can leverage the strengths of MRI whilst allowing for a validation 'in principle' using histology. Furthermore, the same modalities were used in the original human study that we compare to, and establishing a valid comparison was a key aim of this project. The definition of the MRI tissue segmentation and histological subfield labelling in the macaque was supervised by the same researcher, who also performed the manual annotations in the original human BigBrain, thus ensuring maximal consistency. We can, of course, not rule out that the criteria for these labels will be revised in the future, especially due to the recent welcome growth of comparative neuroanatomy studies. However, any changes to the MRI tissue segmentation will have to be implemented for both the macaque and the human segmentation simultaneously, to ensure that the results can still be compared. Changing the criteria for the macaque segmentation only, would effectively invalidate our comparison.

Multi-modal studies that employ comparable criteria across species for both histology and MRI analysis are currently still very rare, and we believe that our study provides an important piece for other researchers to build upon. While the current work visualises our subfield labels in the unfolded sheet-representation of the hippocampus, it would be possible to also represent alternative subfield definitions in both species. With the data and code provided openly by us, other anatomists can easily map their own histology data to this surface and perform other more targeted comparisons, for example by mapping receptor data comparatively in two species.

To address the Reviewer's concern, we now explain the benefits of our framework more explicitly in the Introduction section:

Page 3: "*We capitalise on recent computational approaches to analytically unfold the hippocampal formation, and to derive a surface-based coordinate system³⁰. This topological framework maps the hippocampus intrinsic long (anterior-posterior) and short (distal-proximal) axes, thus respecting the sheet-like anatomy of the hippocampus. Representing the cortex in a surface-based coordinate system has previously proven to advance efforts in brain mapping³¹. Specifically for the hippocampus, such a data-driven estimation of hippocampal coordinates allows us to establish subregional correspondence in a cross-species setting, independent of a specific definition of hippocampal regions or subfields. It*

nevertheless allows for integration of multi-modal data ranging from high-resolution histological data to in-vivo functional MRI in a shared framework.

Furthermore, we acknowledge the important limitation of a manual segmentation now explicitly and we provide the template volumetric MRI tissue segmentation to the Reviewers. Instructions for how to access these data are provided in the first paragraph of our Response letter.

Page 22: *“For further explanations for each MRI label, relations to histology and the criteria used for manual definition, the Reader is referred to the original human protocol¹¹⁰.”*

Page 17: *“While the current study relied on a manual and therefore imperfect macaque MRI tissue segmentation, ongoing developments of the software will further refine the level of detail and range of input modalities and species towards observer-independent and automatic unfolding of the primate hippocampus.”*

Page 30: *“Nissl-stained histology data with digital annotation files, the template MRI tissue segmentation, as well as ex-vivo macaque MRI data are available via WIN’s Digital Brain Bank platform (<https://open.win.ox.ac.uk/DigitalBrainBank³⁵>; dataset title: Hipmac project).”*

R3#3. In the rebuttal letter the authors write that: “The SLRM is drawn within the MRI volume as a relatively ‘thin’ sheet with roughly constant thickness throughout the hippocampus.”. However, the thickness of the radiatum and lacunosum-molecular layers is not constant throughout the hippocampus. It is proportionately thickest in the CA3 region and thinnest in the CA2 region. Furthermore, what the authors have identified in the MRI volume is the so-called “dark band”, which refers to a hypointense line appearing within the grey matter of the hippocampus on high resolution T2 images, and which does not correspond to the entire width of the SRLM. Therefore, throughout the manuscript the authors should refrain from using the term SRLM for their delineations in the MR sequence. Rather, they should simply describe what they have labelled, namely the “so-called dark band or hypointense line appearing within the grey matter of the hippocampus between the pyramidal layer of the CA region and the molecular layer of the dentate gyrus”. This is particularly true for the Material and methods section.

The statement “(SLRM, the ‘hippocampal dark band’ on page 21 must be changed to reflect the following facts: The SLRM is the outer molecular layer of the CA3 region and in MRI images a hypointense line is found roughly at this topographical location.

The statement „SLRM label covers a heterogenous set of structures including the archicortical cortex, but also axons and other incidental structures, such as residual cerebrospinal fluid.” On page 21 must also be changed. „archicortical cortex“ is redundant wording. It is a well established fact that the cortical ribbon also contains axons. Therefore “also axons” is also redundant. The important fact is that the SLRM label covers *part of the hippocampal molecular layer* and also incidental structures, such as residual cerebrospinal fluid. This is precisely why it is not a good idea to call this label SLRM!

We appreciate the Reviewer’s concern with our use of the SLRM terminology. We now follow the Reviewer’s recommendation and use the more descriptive term ‘dark band’ to avoid confusion. To clarify: this dark band is only used by our algorithm to derive a surface that was necessary for the analytical unfolding of the hippocampus. It is not used to segment a cortical layer or other anatomical subdivisions. Nevertheless, and again, to avoid confusion, the revised version of the manuscript refrains from using the term SLRM and instead calls it ‘dark band’. Accordingly, we’ve adapted the description of the label and added clarifications as described above for R3#2:

Page 22: “The ‘dark band’ label covers a heterogenous set of structures including parts of the archicortical strata radiatum, lacunosum and moleculare, but also other axons and incidental structures, such as residual cerebrospinal fluid and blood vessels in the hippocampal sulcus.”

R3#4. It is worrying that the authors consider that the labels needed to guide the unfolding algorithm of HipUnfold are of no interest anatomically, since (if I understood correctly) the tool was developed to provide the framework for the analysis of anatomically relevant scientific questions.

We thank the Reviewer for raising this point and, of course, these labels are anatomically relevant for our framework. In order to unfold the hippocampus, the algorithm uses the information about where the labels from the MRI tissue segmentation touch each other, rather than their full volumetric extent. For example, if the indusium griseum label extends more or less posteriorly is not relevant for the output of the software. An analogy might be helpful here: In the field of MRI, many analyses begin with brain extraction, or skull stripping. An algorithm might use the term ‘skull’ for non-brain voxels, even if these voxels contain other anatomical structures such as the dura mater or the meninges. This does not invalidate the brain extraction algorithm.

To avoid confusions, we introduced a new paragraph in the Method section that summarises our analysis framework, prior to detailing the different steps (Page 21):

“Overview of hippocampal unfolding approach

First, we conducted an MRI tissue segmentation to identify robust landmarks of the hippocampus and surroundings. In the macaque we defined the segmentation manually, whilst in the human the corresponding segmentation was derived automatically. Next, the software tool hippunfold³⁰ was used to estimate hippocampal coordinates along the short and long hippocampal axis based on this segmentation. The tool further reconstructs the hippocampal surface and computes a coordinate transformation to achieve analytical unfolding or flattening of the hippocampus. This ultimately results in a surface-based coordinate system that is matched across the two species. Hippocampal subfield labels were manually defined in the macaque histology data, then translated to MRI-space and finally sampled along the hippocampal surface, resulting in a 2D map. In the human, the corresponding map of subfields was accessed from a previous study¹⁰⁵.”

R3#5. Supplementary file 1 is a nice figure for a presentation, but of little scientific use, as it does not enable assessment of the correctness of anatomical labels.

The purpose of this supplementary figure is to enable the reader to assess the spatial continuity and topological relationship of hippocampal subfields in a 3D perspective, complementing representations of consecutive histology slices. We now show a high-resolution display of the histology slices as well, see our comment below (R3#6).

R3#6. Supplementary figure 4 only shows four sections and the resolution of the images provided is not sufficient to enable assessment, e.g., of the correctness of the CA3 label. I can only repeat my request. The authors should provide a figure demonstrating, for the entire rostro-caudal extent of the hippocampus (and at an adequate resolution), their manual segmentations. Alternatively, a part of the review process the reviewers should be able to access the Im 83 histology slices that the authors plan to make available upon publication.

Previously, we labelled the subfields in the medium-resolution slices available to us and validated our labelling strategy by cross-checking with a subset of interleaved high-resolution slices. To accommodate the concern of the Reviewer, however, our collaborators now kindly made all raw high-resolution slices available, and we re-drew all histological labels at maximal resolution (0.28 µm/pixel) with minimal corrections. In the revised manuscript, Supplementary Figure S4 is now replaced with a

Supporting Data file, shows all available slices in both hemispheres at higher resolution. As the file-size exceeds that of a regular manuscript supplement, we uploaded it via the Figshare tool provided by the journal and is available at [REDACTED]

The digital resolution in this file is inherently limited by the pixel resolution of the pdf format. Therefore, we also provide access to the high-resolution slides (in tif format) and their digital annotation files (in geojson format), which can be loaded in the QuPath image viewer software. These slices are cropped to show only the hippocampus with surroundings, and they are minimally down-sampled by factor 10 (2.8 $\mu\text{m}/\text{pixel}$) to keep within a reasonable folder size. Instructions for how the Reviewers can access these data are provided in the first paragraph of our Response letter.

To reflect these changes, we have adapted the manuscript accordingly:

Page 23: “Hippocampal subfields were manually labelled in QuPath v.0.2.3¹⁰⁷ onto the Cresyl Violet stained histological slices at a resolution of 0.28 μm / pixel (see **Figure 1D** for an example and **Supporting Data on Figshare** for all slices). All histology data with annotations are provided open access (see Data Availability section).”

Page 30: “Supporting Data with screenshots of the hippocampal sections with annotations are openly available on Figshare at: [REDACTED]

Examples are shown below:

An example screenshot of a labelled histology slice as provided in the Supporting Data. In total 83 slices from both hemispheres are shown in the PDF, ordered by approximate y-coordinate from posterior to anterior.

An example high-resolution histology file shown with the digital annotation file overlaid as visualised in QuPath. All 83 histology slices with annotation files have been uploaded to the OSF repository, which is accessible to the Reviewers.

A zoomed-in section of the histology slice above demonstrates the high resolution of the provided data, which allows for identification of stained cells.

[REDACTED]

Thumbnail of instructions on the Wiki-page of the OSF repository shared with the Reviewers.

R3#7. Even if the parcellation is not relevant for the connectivity analysis, it must be controlled from the anatomical point of view

We agree with the Reviewer, of course, that the hippocampal subfield segmentation should be in agreement with the anatomical literature. Our main analyses are based on the unfolded surface space, defined by topological estimation of hippocampal coordinates along the short and long hippocampal axis. The histology-based hippocampal subfield labels are sampled along this surface to provide context and as intuitive visualisation of the rectangular 2D flatmap. Therefore, minor deviations in subfield boundaries do not represent a major confound for our downstream results and conclusions. We explicitly state the limitation of our approach in the Discussion section

Page 14: *“Although findings on potential cross-species microstructural differences require further validation in a larger histological sample to discern inter-species from inter-individual differences, the*

methodology we introduced here offers a scalable framework to allow for microstructural and subregional comparisons across humans and non-human primates.”

Together with additional acknowledgements of limitations introduced in this Revision (see in particular R3#2), we hope this addresses the concern of the Reviewer.

R3#8. Minor points:

- The authors should label which part of Fig. 1E shows the macaque (BigMac) hippocampal subfields and which the human (BigBrain) subfields.

As suggested, we revised the label of Fig. 1E by replicating the label from Fig. 1B above.

- In Fig. 1E the authors have labelled (among others) the CA1 and pre-/subiculum regions. The prosubiculum is found between the CA1 region and the Subiculum. Is this area encompassed by the CA1 label or by the pre-/subiculum label?

The pre-/subiculum also encompasses the prosubiculum. To avoid confusion, we decided to refrain from the term pre-/subiculum in the manuscript and instead call it ‘subicular complex’.

- What is the meaning of the abbreviation a.u. in Fig. 1C?

It stands for ‘arbitrary unit’ as normalised intensities are shown. All abbreviations used in the figures are now introduced in the figure captions.

REVIEWERS' COMMENTS

Reviewer #2 (Remarks to the Author):

Reviewer #3 (Remarks to the Author):

The authors have adressed all my concerns